# Knowing Your Target 🎯: Target-Aware Transformer Makes Better Spatio-Temporal Video Grounding

**Xin Gu**[1]    **Yaojie Shen**[2,1]    **Chenxi Luo**[3]    **Tiejian Luo**[1,*]    **Yan Huang**[4]    **Yuewei Lin**[5]
**Heng Fan**[4,†]    **Libo Zhang**[2,1,†,*]
[1]University of Chinese Academy of Sciences
[2]Institute of Software Chinese Academy of Sciences    [3]La Trobe University
[4]University of North Texas    [5]Brookhaven National Laboratory
guxin21@mails.ucas.edu.cn, libo@iscas.ac.cn

## Abstract

Transformer has attracted increasing interest in spatio-temporal video grounding, or STVG, owing to its end-to-end pipeline and promising result. Existing Transformer-based STVG approaches often leverage a set of object queries, which are initialized simply using zeros and then gradually learn target position information via iterative interactions with multimodal features, for spatial and temporal localization. Despite simplicity, these zero object queries, due to lacking target-specific cues, are hard to learn discriminative target information from interactions with multimodal features in complicated scenarios (*e.g.*, with distractors or occlusion), resulting in degradation. Addressing this, we introduce a novel *Target-Aware Transformer for STVG* (**TA-STVG**), which seeks to adaptively generate object queries via exploring target-specific cues from the given video-text pair, for improving STVG. The key lies in two simple yet effective modules, comprising *text-guided temporal sampling* (TTS) and *attribute-aware spatial activation* (ASA), working in a cascade. The former focuses on selecting target-relevant temporal cues from a video utilizing holistic text information, while the latter aims at further exploiting the fine-grained visual attribute information of the object from previous target-aware temporal cues, which is applied for object query initialization. Compared to existing methods leveraging zero-initialized queries, object queries in our TA-STVG, directly generated from a given video-text pair, naturally carry target-specific cues, making them adaptive and better interact with multimodal features for learning more discriminative information to improve STVG. In our experiments on three benchmarks, including HCSTVG-v1/-v2 and VidSTG, TA-STVG achieves state-of-the-art performance and significantly outperforms the baseline, validating its efficacy. Moreover, TTS and ASA are designed for general purpose. When applied to existing methods such as TubeDETR and STCAT, we show substantial performance gains, verifying its generality. Code is released at https://github.com/HengLan/TA-STVG.

## 1 Introduction

Spatio-temporal video grounding (or **STVG**) aims at *spatially* and *temporally* localizing the target of interest from an untrimmed video given its textual description (Zhang et al., 2020b). As a multimodal task, it requires to understand the spatio-temporal content from videos and accurately connect it to the corresponding textual expression. Owing to its importance in multimodal video understanding and key applications such as content-based video retrieval, robotics, etc, STVG has attracted increasing interest with many models proposed (Zhang et al., 2020a;b; Tang et al., 2021; Su et al., 2021).

Recently, inspired by the compact end-to-end pipeline of Transformer-based detection (Carion et al., 2020), researchers have introduced the Transformer (Vaswani et al., 2017) for STVG given that they both are localization task, and achieved previously unattainable result (Yang et al., 2022a; Jin et al.,

---

[†]Equal advising and co-last authors; [*]Corresponding author

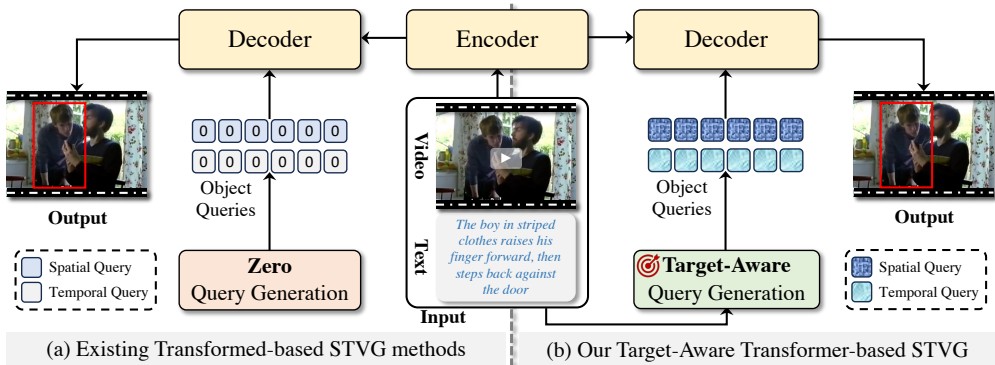

Figure 1: Comparison between existing Transformer-based STVG methods applying zero-initialized queries for STVG in (a) and our proposed Target-Aware Transformer-based STVG generating queries with target-aware cues from video and text for STVG in (b). *Best viewed in color for all figures*.

2022; Lin et al., 2023b; Gu et al., 2024a). Similar to Transformer-based detection (Carion et al., 2020), existing Transformer-based STVG methods often adopt a set of spatial and temporal object queries, and then leverage them to learn target position information by iterative interactions with multimodal features (generated by the encoder) from video and text for spatio-temporal target localization. In these approaches (Yang et al., 2022a; Jin et al., 2022; Lin et al., 2023b; Gu et al., 2024a), the object queries are usually simply initialized using *zeros* (see Fig. 1 (a)). Despite promising results, such zero-initialized object queries, due to the lack of effective target-specific semantic cues, are difficult to learn discriminative target position information from the interactions with multimodal features in complicated scenarios, *e.g.*, with distractors or occlusion, leading to performance degradation.

In the decoder procedure, object queries expect to learn target position information from multimodal features. If *object queries know the target from the very beginning*, or in other words, *they know what to learn*, they can employ target-specific cues as a prior to guide themselves for better interaction with the multimodal features, which benefits learning more discriminative features for better localization. In order to validate this, we conduct the *oracle* experiment by generating spatial and temporal object queries from groundtruth, and compare the performance using such groundtruth-generated queries and zero-initialized queries. The architecture in this oracle experiment simply consists of feature extraction, an encoder and a decoder, and please refer to the supplementary material for details due to limited space. Fig. 2 shows the comparison results

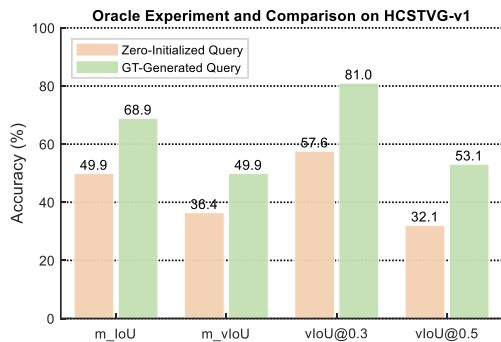

Figure 2: Comparison of the zero-initialized queries and groundtruth-generated queries for STVG. We see the target-specific information in groundtruth largely enhances results.

on HCSTVG-v1 (Tang et al., 2021), and more comparison on other benchmarks can be found in Sec. A in the supplementary material. From Fig. 2, we can clearly observe that, when introducing the target-specific information from groundtruth to initialize object queries, the STVG performance can be significantly improved, *e.g.*, from 49.9% to 68.9% in m_IoU (19.0% absolute gains) and 36.4% to 49.9% in m_vIoU (13.5% absolute gains), fully verifying effectiveness of target-specific information in initializing queries for accurate STVG.

Thus motivated by the above, we introduce a novel *Target-Aware Transformer for STVG* (TA-STVG), which seeks to adaptively initialize object queries by exploring target-specific cues from videos for spatio-temporal target localization (see Fig. 1 (b)). The crux of our TA-STVG lies in two simple yet effective modules, including *text-guided temporal sampling* (TTS) and *attribute-aware spatial activation* (ASA). These two modules work in a cascade in the TA-STVG, from coarse to fine, to generate the target-aware queries for subsequent localization. Specifically, TTS, from the temporal perspective, firstly aims to select target-relevant frames from the video using holistic information from the textual expression. To enhance the temporal selection, both appearance and motion cues are jointly considered in TTS. Then, from a spatial perspective, ASA focuses on exploring fine-grained

visual semantic information of the object from the previous target-aware temporal cues learned by TTS. Particularly, attributes such as color and action will be exploited in the ASA to activate spatial features most relevant to the target object. Considering the specificity of object queries for spatial and temporal decoders, different activation features based on the specific attributes will be generated. Finally, features from ASA are used to initialize queries, which are fed to decoders, interacting with multimodal features from the encoder, to learn target position information for localization.

In this work, we employ the DETR-similar architecture for TA-STVG, similar to existing Transformer-based STVG methods (Yang et al., 2022a; Jin et al., 2022; Gu et al., 2024a), but **difference** is that, our object queries, for both spatial and temporal localization, are directly generated from the given video-text pair, and thus naturally carry the target-specific cues, making them adaptive and better interact with multimodal features for learning more discriminative information to improve STVG. Our attribute-aware activation is related to Lin et al. (2023b) that activates the spatially attended target using intra-frame visual cues and  Tan et al. (2024) that jointly learns feature alignment and regression using weak supervision signals. The **difference** is that, our ASA mines target-relevant spatial cues from features by TTS for learning target-aware queries to enhance target localization. To our best knowledge, TA-STVG is the first to incorporate target-specific cues for object query generation to improve STVG. To validate its efficacy, we conduct experiments on HCSTVG-v1/-v2 (Tang et al., 2021) and VidSTG (Zhang et al., 2020b), and results show that TA-STVG outperforms previous STVG methods by achieving new state-of-the-arts, evidencing effectiveness of our solution. Moreover, our TTS and ASA are designed for general purpose and thus applicable to other architectures. We apply TTS and ASA on top of two approaches, including the seminal Transformer-based TubeDETR (Yang et al., 2022a) and STCAT (Jin et al., 2022), significantly improving their performance, as shown in our experiments, further highlighting the generality of our approach.

In summary, the major contributions of this work are as follows: ♠ We present the TA-STVG, a novel Target-Aware Transformer for improving STVG by exploring target-specific cues for object queries; ♥ We propose text-guided temporal sampling (TTS) for selecting target-relevant temporal cues from the videos; ♣ We present attribute-aware spatial activation (ASA) to exploit fine-grained visual semantic attribute information for object query generation; and ♦ TA-STVG in extensive experiments achieves new state-of-the-art performance and shows good generality, demonstrating its efficacy.

## 2 RELATED WORK

**Spatio-Temporal Video Grounding.** Spatio-temporal video grounding (STVG) (Zhang et al., 2020b) aims to spatially and temporally localize the target of interest, given its free-form of textual description, from an untrimmed sequence. Early STVG approaches (Zhang et al., 2020b;a; Tang et al., 2021) mainly consists of two stages, *first* generating candidate proposals from the video with a pre-trained detector and *then* selecting correct proposals based on the textual expression. To eliminate the heavy dependency on the pre-trained detection model, recent methods (Su et al., 2021; Yang et al., 2022a; Jin et al., 2022; Lin et al., 2023b; Talal Wasim et al., 2024; Gu et al., 2024a), inspired by Transformer, switch to the one-stage pipeline that directly generates a spatio-temporal tube for target localization, without relying on any external detectors. Owing to the compact end-to-end training pipeline, such a one-stage framework demonstrates superior performance compared to previous two-stage algorithms. Our TA-STVG also belongs to the one-stage Transformer-based type. However, **different from** the aforementioned Transformer-based approaches that simply follows (Carion et al., 2020) to leverage zero-initialized object queries for target localization, TA-STVG innovatively exploits target-specific cues from the video-text pair for object query generation, making it adaptive to various scenarios and better interact with multimodal features in the decoder for more accurate localization.

**Temporal Grounding.** Temporal grounding focuses on localizing specific targets or events from the video given the textual expression. Being relevant to STVG, it requires temporally localizing the target of interest, but the difference is that temporal grounding does not require to perform the spatial bounding box localization. In recent years, many methods (Cao et al., 2022; Chen et al., 2021; Mun et al., 2020; Hao et al., 2022; Wang et al., 2023b; Zhang et al., 2023; Barrios et al., 2023; Lin et al., 2023a) have been proposed for temporal grounding. For instance, the work of (Wang et al., 2023b) introduces a pre-training approach for improving video temporal grounding. The method in (Lin et al., 2023a) presents a unified framework for various video temporal grounding tasks. The approach in (Chen et al., 2021) proposes to learn complementary features from different modalities including

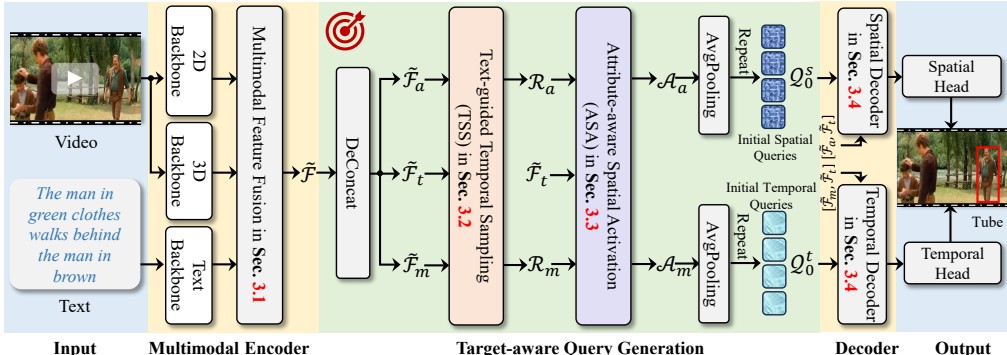

Figure 3: Overview of TA-STVG, which exploits target-specific information from the video and text (*i.e.*, features from multimodal encoder) for generating spatial and temporal object queries for STVG.

images, flow, and depth for temporal grounding. ***Different than*** these works, we focus on the more challenging STVG involved with spatial and temporal localization of the target.

**Transformer-based Detection.** Detection is a fundamental component in computer vision. Recently, the seminal work DETR (Carion et al., 2020) has applied Transformer (Vaswani et al., 2017) for detection with impressive performance, and later been further improved in numerous extensions (Sun et al., 2021; Zhu et al., 2021; Zheng et al., 2023; Ye et al., 2023). Similar to other Transformer-based STVG works, our method employs the DETR-similar architecture for STVG. The ***difference*** is that we propose to learn adaptive object queries from the video-text pair, instead of utilizing zero object queries following (Carion et al., 2020) as in existing STVG methods, for better target localization.

**Vision-Language Modeling.** Vision-language modeling (VLM) aims to process both visual content and language for multimodal understanding. In recent years, it has attracted great attention from the researchers and been studied in various tasks including visual question answering (Antol et al., 2015; Jiang et al., 2020; Han et al., 2023; Shao et al., 2023; Wang et al., 2024b; Weng et al., 2024), visual captioning (You et al., 2016; Gu et al., 2023; 2022; Wang et al., 2024a; Shen et al., 2023; Ren et al., 2024; Wang et al., 2023a), navigation (Zhu et al., 2020; Li & Bansal, 2023), text-to-image generation (Gu et al., 2024b; Ramesh et al., 2021), referring expression segmentation (Yang et al., 2022b; Liu et al., 2023), vision-language tracking (Guo et al., 2022; Zhou et al., 2023), etc. ***Different*** from these tasks, we focus on vision-language modeling for spatio-temporal video grounding.

## 3 OUR APPROACH

**Overview.** In this work, we propose TA-STVG, which aims to generate target-aware object queries for improving STVG. Motivated by DETR (Carion et al., 2020), TA-STVG utilizes the encoder-decoder framework which consists of a multimodal encoder and a spatio-temporal grounding decoder. As shown in Fig. 3, the multimodal encoder (§3.1) extracts and interacts visual and textual features, while the decoder (§3.4) learns target position information using target-aware object queries generated with TTS (§3.2) and ASA (§3.3) and their interaction with the multimodal feature for target localization.

### 3.1 MULTIMODAL ENCODER

Given a video and the textual expression, the multimodal encoder aims at obtaining vision-language features for target-aware query generation and the subsequent decoder for localization. It comprises visual and textual feature extraction and multimodal feature fusion as described in the following.

**Visual and Textual Feature Extraction.** For the video, we extract its both appearance and motion features to leverage rich static and dynamic information. Specifically, we first sample $N_v$ frames $\mathcal{I} = \{I_i\}_{i=1}^{N_v}$ from the video. Afterwards, ResNet-101 (He et al., 2016) and VidSwin (Liu et al., 2022) are respectively utilized for appearance and motion feature extraction. The appearance feature is denoted as $\mathcal{F}_a = \{f_i^a\}_{i=1}^{N_v}$, where $f_i^a \in \mathbb{R}^{H \times W \times C_a}$ with $H$, $W$ and $C_a$ the height, width and channel dimensions, and the motion feature is represented using $\mathcal{F}_m = \{f_i^m\}_{i=1}^{N_v}$, where $f_i^m \in \mathbb{R}^{H \times W \times C_m}$ with $C_m$ the channel dimension.

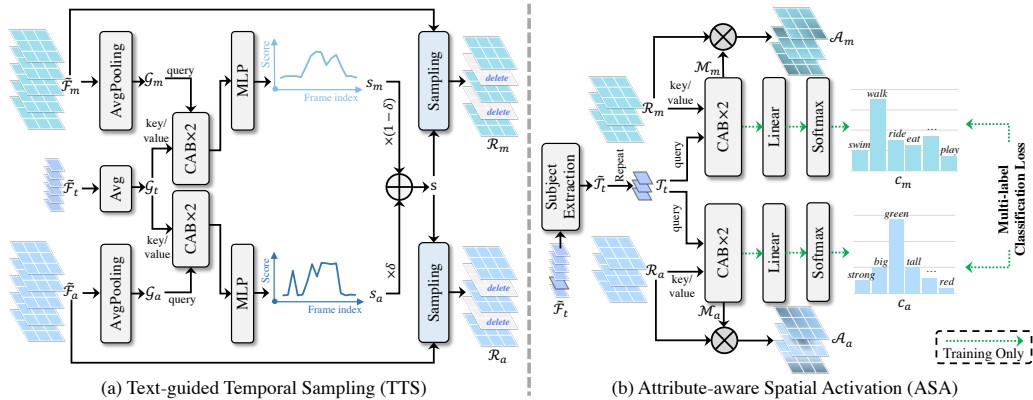

Figure 4: Illustration of the architecture for TTS in (a) and ASA in (b).

For the textual expression, we use RoBERTa (Liu et al., 2019) to extract its feature. We first tokenize it to obtain a word sequence $\mathcal{W} = \{w_i\}_{i=1}^{N_t}$ containing $N_t$ words. Then, we apply RoBERTa on $\mathcal{W}$ to generate the textual feature $\mathcal{F}_t = \{f_i^t\}_{i=1}^{N_t}$, where $f_i^t \in \mathbb{R}^{C_t}$ with $C_t$ the textual feature channel.

**Multimodal Feature Fusion.** For enhancing feature representation for STVG, we fuse multimodal features of appearance feature $\mathcal{F}_a$, motion feature $\mathcal{F}_m$, and text feature $\mathcal{F}_t$, similar to (Gu et al., 2024a). Specifically, we first project them to the same channel number $D$ and then concatenate corresponding features to produce the multimodal feature $\mathcal{F} = \{f_i\}_{i=1}^{N_v}$ as follows,

$$f_i = [\underbrace{f_{i_1}^a, f_{i_2}^a, ..., f_{i_{H \times W}}^a}_{\text{appearance features } f_i^a}, \underbrace{f_{i_1}^m, f_{i_2}^m, ..., f_{i_{H \times W}}^m}_{\text{motion features } f_i^m}, \underbrace{f_1^t, f_2^t, ..., f_{N_t}^t}_{\text{textual features } \mathcal{F}_t}] \tag{1}$$

where $f_i$ is the multimodal feature in frame $i$. Then, we perform feature fusion using the self-attention encoder (Vaswani et al., 2017) to obtain the multimodal feature $\tilde{\mathcal{F}}$ as follows,

$$\tilde{\mathcal{F}} = \texttt{SelfAttEncoder}(\mathcal{F} + \mathcal{E}_{pos} + \mathcal{E}_{typ}) \tag{2}$$

where $\mathcal{E}_{pos}$ and $\mathcal{E}_{typ}$ denote position and type embeddings imposed on $\mathcal{F}$, and $\texttt{SelfAttEncoder}(\cdot)$ is the self-attention encoder with $L$ ($L$=6) standard self-attention encoder blocks. Please refer to Sec. B in supplementary material for architecture of $\texttt{SelfAttEncoder}(\cdot)$.

### 3.2 TEXT-GUIDED TEMPORAL SAMPLING (TTS)

To generate target-specific object queries, we first develop a *text-guided temporal sampling* (TTS) module to identify and sample frames relevant to the target guided by holistic textual features. Specifically, it will predict a relevance score between each frame and text and then sample video frames with their relevance scores above a predefined threshold. To enhance the frame selection in TTS, both appearance and motion are considered in computing the relevance, as shown in Fig. 4 (a).

More concretely, given multimodal feature $\tilde{\mathcal{F}}$ from the multimodal encoder (§3.1), we first extract the appearance, motion and textual features, respectively denoted by $\tilde{\mathcal{F}}_a$, $\tilde{\mathcal{F}}_m$, and $\tilde{\mathcal{F}}_t$, from $\tilde{\mathcal{F}}$ via deconcatenation $[\tilde{\mathcal{F}}_a, \tilde{\mathcal{F}}_m, \tilde{\mathcal{F}}_t] = \texttt{DeConcat}(\tilde{\mathcal{F}})$. Afterwards, we perform average pooling on $\tilde{\mathcal{F}}_a$ and $\tilde{\mathcal{F}}_m$ and channel-average operation on $\tilde{\mathcal{F}}_t$ to obtain frame-level appearance and motion features and the holistic textual feature,

$$\mathcal{G}_a = \texttt{AvgPooling}(\tilde{\mathcal{F}}_a) \qquad \mathcal{G}_m = \texttt{AvgPooling}(\tilde{\mathcal{F}}_m) \qquad \mathcal{G}_t = \texttt{Avg}(\tilde{\mathcal{F}}_t) \tag{3}$$

where $\mathcal{G}_a$, $\mathcal{G}_m$ (both $\in \mathbb{R}^{N_v \times D}$) and $\mathcal{G}_t$ ($\in \mathbb{R}^{1 \times D}$) are averaged appearance, motion, and text features.

To measure the relevance of each frame with the text, we fuse the textual feature into appearance and motion features via two cross-attention blocks, respectively. Specifically, the appearance feature $\mathcal{G}_a$ or motion feature $\mathcal{G}_m$ serves as query, while the textual feature $\mathcal{G}_t$ as key/value. After this, MLP is performed on the appearance and motion features to predict relevance scores for each frame in terms of appearance and motion. As shown in Fig. 4 (a), this process can be expressed as follows,

$$s_a = \texttt{MLP}(\texttt{CA}(\texttt{CA}(\mathcal{G}_a, \mathcal{G}_t), \mathcal{G}_t)) \qquad s_m = \texttt{MLP}(\texttt{CA}(\texttt{CA}(\mathcal{G}_m, \mathcal{G}_t), \mathcal{G}_t)) \tag{4}$$

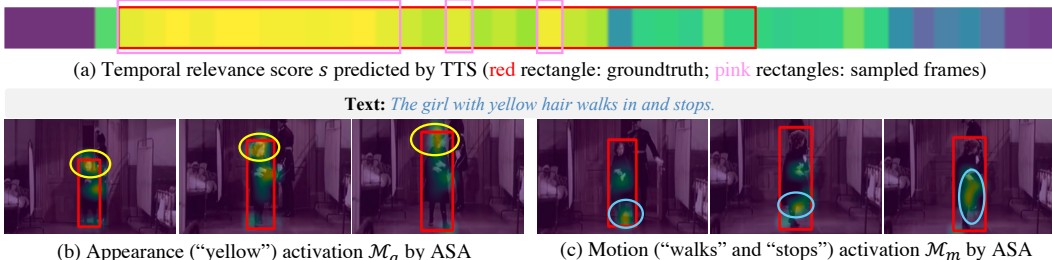

(a) Temporal relevance score $s$ predicted by TTS (red rectangle: groundtruth; pink rectangles: sampled frames)

**Text:** *The girl with yellow hair walks in and stops.*

(b) Appearance ("yellow") activation $\mathcal{M}_a$ by ASA      (c) Motion ("walks" and "stops") activation $\mathcal{M}_m$ by ASA

Figure 5: Illustration of temporal relevance score $s$ by TTS in (a) and attribution-aware spatial activation (in partial selected frames) including appearance and motion activation in (b) and (c). We can see from (a) that TTS can accurately select target-relevant frames, and from (b) and (c) that ASA precisely localizes attributes, *e.g.*, color "*yellow*" and action "*walks in, stops*", related to the target.

where $s_a$ and $s_m$ are the relevance scores regarding appearance and motion, $\texttt{CA}(\mathbf{z}, \mathbf{u})$ denotes the cross-attention block (CAB) (Vaswani et al., 2017) with $\mathbf{z}$ generating query and $\mathbf{u}$ key/value, and $\texttt{MLP}(\cdot)$ is the MLP.

Considering both appearance and motion are critical when determining target-relevant frames, we combine $s_a$ and $s_m$, controlled by $\delta$, to compute the final relevance score $s$ for each frame, as follows,

$$s = \delta \times s_a + (1 - \delta) \times s_m \tag{5}$$

With $s$, we leverage it to sample target-relevant appearance and motion features. Specifically, we only keep appearance and motion features in frames whose relevance scores are above the pre-defined threshold $\theta$, and this process can be written as follows,

$$\begin{aligned} \mathcal{R}_a &= \texttt{sample}(\tilde{\mathcal{F}}_a, s, \theta) = \{\tilde{\mathcal{F}}_a(i) | s(i) > \theta\} \\ \mathcal{R}_m &= \texttt{sample}(\tilde{\mathcal{F}}_m, s, \theta) = \{\tilde{\mathcal{F}}_m(i) | s(i) > \theta\} \end{aligned} \tag{6}$$

where $i \in [1, N_v]$ is the frame index. $\mathcal{R}_a$ and $\mathcal{R}_m$ are sampled target-relevant temporal appearance and motion features, which will be used for extracting target-specific fine-grained attribute information. Fig. 5 (a) shows an example of the temporal relevance score and selected temporal frames by TTS. Please note, for $\mathcal{R}_a$ and $\mathcal{R}_m$, their quantities are the same, because we set the same threshold $\theta$ for both appearance and motion features, and sample $\mathcal{R}_a$ and $\mathcal{R}_m$ from $\tilde{\mathcal{F}}_a$ and $\tilde{\mathcal{F}}_m$ ($\tilde{\mathcal{F}}_a$ and $\tilde{\mathcal{F}}_m$ have the same quantities) both based on the relevance score $s$ generated by TTS (see Eq. (6)). Thus, their quantities are the same, and consistent with the number of frames sampled by TTS.

### 3.3 ATTRIBUTE-AWARE SPATIAL ACTIVATION (ASA)

To explore more fine-grained information of the target, we design an *attribute-aware spatial activation* (ASA) module, which mines fine-grained visual semantic information from previous coarse-grained temporal features $\mathcal{R}_a$ and $\mathcal{R}_m$ for object queries. Particularly, ASA considers appearance attribute like color (*e.g.*, "*green*", "*red*", etc) and shape (*e.g.*, "*small*", "*tall*", etc) and motion attribute such as action (*e.g.*, "*walk*", "*ride*", etc) to generate fine-grained spatial features for the target. The ***intuition*** is that, appearance attribute is important for spatial target localization, while motion attribute is crucial in temporal localization, *i.e.*, locating the start and end of the tube. In certain cases, even if spatial localization of target is good enough in each frame, the accuracy is low due to the incapability of temporally recognizing target action. Considering the specificity of queries for spatial and temporal decoders, appearance and motion activation features based on specific attributes are generated.

Specifically, as shown in Fig. 4 (b), given the textual feature $\tilde{\mathcal{F}}_t$, we first extract the feature $\tilde{\mathcal{T}}_t$ of the subject (usually the target to localize) in the textual sentence from it. Due to limited space, please refer to the Sec. D in supplementary material for subject extraction. Then, we repeat the subject feature $\tilde{\mathcal{T}}_t$ to match the number of frames in $\mathcal{R}_a$ and $\mathcal{R}_m$, and the resulted feature is denoted as $\mathcal{T}_t$. After that, we learn the fine-grained spatial features from $\mathcal{R}_a$ and $\mathcal{R}_m$ by interacting them with $\mathcal{T}_t$ respectively using two cross-attention blocks. In specific, the subject feature $\mathcal{T}_t$ serves as the query, while $\mathcal{R}_a$ and $\mathcal{R}_m$ as the key/value. By doing so, we can allow the subject feature to learn relevant spatial features from appearance and motion. To enforce the learning of attribute-aware

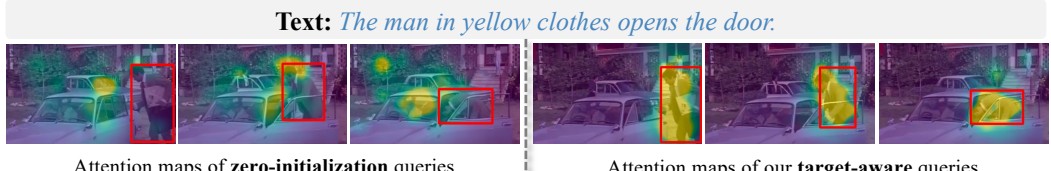

Figure 6: Comparison of attention maps for zero-initialized object queries and the proposed target-aware object queries in video frames in the spatial decoder. The red boxes indicate the foreground target to localize. From this figure, we can clearly observe that the attention maps of our target-aware object queries by TTS and ASA can better focus on the target object for localization.

spatial features, we propose to supervise ASA with *explicit* weak attribute labels generated from the textual expression using multi-label classification losses for appearance and motion. The appearance and motion are multi-class labels. Due to space limitation, we refer readers to our supplementary material for details regrading the attribute label construction. The attribute classification is performed with a linear projection layer followed with softmax. As in Fig. 4 (b), this can be written as follows,

$$c_a = \texttt{Softmax}(\texttt{Linear}(\texttt{CA}(\texttt{CA}(\mathcal{T}_t, \mathcal{R}_a), \mathcal{R}_a)))$$
$$c_m = \texttt{Softmax}(\texttt{Linear}(\texttt{CA}(\texttt{CA}(\mathcal{T}_t, \mathcal{R}_m), \mathcal{R}_m)))$$

$$(7)$$

where $c_a$ and $c_m$ represent classification scores for appearance and motion attributes, respectively, and $\texttt{Linear}(\cdot)$ and $\texttt{Softmax}(\cdot)$ denote linear projection and softmax classification.

By supervising $c_a$ and $c_m$ with appearance and motion attribute labels, the subject feature $\mathcal{T}_t$, serving as query in cross-attention blocks, is able to adaptively learn fine-grained features relevant to attributes from $\mathcal{R}_a$ and $\mathcal{R}_m$ for attribute classification. Therefore, the attention maps of $\mathcal{T}_t$ with $\mathcal{R}_a$ and $\mathcal{R}_m$ can be explained as attribute-specific spatial activation in appearance and motion (Dosovitskiy et al., 2021), and adopted to generate attribute features from $\mathcal{R}_a$ and $\mathcal{R}_m$ as follows,

$$\mathcal{A}_a = \mathcal{M}_a \otimes \mathcal{R}_a \qquad \mathcal{A}_m = \mathcal{M}_m \otimes \mathcal{R}_m \qquad (8)$$

where $\mathcal{A}_a$ and $\mathcal{A}_m$ are the target-specific attribute features and $\mathcal{M}_a$ and $\mathcal{M}_m$ the attribute-aware spatial activation in appearance and motion. Fig. 5 (b) and (c) show the learned attribute activation.

***Discussion.*** Please note that, besides our method, another way is to directly learn spatial activation of the target (*i.e.*, instance-level activation, *not* the attribute-specific activation) supervised by binary masks generated from the groundtruth box. In this method, results after softmax in ASA will be directly employed as the appearance and motion spatial activation for feature extraction (see details in Sec. C in supplementary material). Despite simplicity, this will *not* enable the learning of fine-grained unique attribute features, which are discriminative to distinguish the target from background. As shown in an experiment later, our strategy of ASA shows better performance.

**Generating Object Queries.** After the TTS and ASA, the final fine-grained target-specific features $\mathcal{A}_a$ and $\mathcal{A}_m$ are used to generate the spatial and temporal object queries. Specifically, we simply use average pooling on $\mathcal{A}_a$ and $\mathcal{A}_m$ (note, reshaping is needed) and then apply the repeat operation to obtain the initial spatial and temporal object queries $\mathcal{Q}_0^s$ and $\mathcal{Q}_0^t$ as follows,

$$\mathcal{Q}_0^s = \texttt{repeat}(\texttt{AvgPooling}(\mathcal{A}_a), N_v) \qquad \mathcal{Q}_0^t = \texttt{repeat}(\texttt{AvgPooling}(\mathcal{A}_m), N_v) \qquad (9)$$

where $\texttt{repeat}(\mathbf{v}, k)$ denotes repeating $\mathbf{v}$ by $k$ times. Afterwards, $\mathcal{Q}_0^s$ and $\mathcal{Q}_0^t$ are sent to the decoder for spatial-temporal target localization as described later.

### 3.4 DECODER FOR SPATIO-TEMPORAL LOCALIZATION

To learn target position information, we employ spatial and temporal decoders, similar to existing methods (Jin et al., 2022; Gu et al., 2024a). In both decoders, object queries $\mathcal{Q}_0^s$ and $\mathcal{Q}_0^t$ will iteratively interact with the multimodal feature $\tilde{\mathcal{F}}$ from the encoder for target information learning, as follows,

$$\mathcal{Q}_K^s = \texttt{SpatialDecoder}(\mathcal{Q}_0^s, \tilde{\mathcal{F}}_a, \tilde{\mathcal{F}}_t) \qquad \mathcal{Q}_K^t = \texttt{TemporalDecoder}(\mathcal{Q}_0^t, \tilde{\mathcal{F}}_m, \tilde{\mathcal{F}}_t) \qquad (10)$$

where $\mathcal{Q}_K^s$ and $\mathcal{Q}_K^t$ are learned spatial and temporal queries after decoders, and $\texttt{SpatialDecoder}(\cdot, \cdot)$ and $\texttt{TemporalDecoder}(\cdot, \cdot)$ denote spatial and temporal decoders with each containing $K$ ($K = 6$)

self-attention blocks and cross-attention blocks. Fig. 6 shows the attention maps of our target-aware queries for localization. Due to limited space, please refer to the Sec. B in supplementary material for decoder architectures and additional attention maps of our target-aware queries for localization.

Once generating $\mathcal{Q}_K^s$ and $\mathcal{Q}_K^t$, we employ two heads to predict the final object boxes $\mathcal{B} = \{b_i\}_{i=1}^{N_v}$ where $b_i \in R^4$ represents central position, width and height of predicted box in frame $i$, and start and end probabilities of each frame $\mathcal{H} = \{(h_i^s, h_i^e)\}_{i=1}^{N_v}$, where start and end times are determined by the maximum joint start and end probability.

### 3.5 OPTIMIZATION

In TA-STVG, we predict: (1) the temporal relevance scores $s_a$ and $s_m$ in terms of appearance and motion in TTS, (2) the appearance and motion attribution classification scores $c_a$ and $c_m$ in ASA, and (3) the spatial boxes $\mathcal{B} = \{b_i\}_{i=1}^{N_v}$ in the spatial decoder, start timestamps $\mathcal{H}_s = \{h_i^s\}_{i=1}^{N_v}$ and end timestamps $\mathcal{H}_e = \{h_i^e\}_{i=1}^{N_v}$ in the temporal decoder. During training, with the groundtruth of motion attribute $c_m^*$, appearance attribute $c_a^*$, start timestamps $\mathcal{H}_s^*$, end timestamps $\mathcal{H}_e^*$, and the bounding box $\mathcal{B}^*$, we can calculate the total loss $\mathcal{L}$ as

$$\mathcal{L} = \underbrace{\lambda_{\text{TTS}}(\mathcal{L}_{\text{BCE}}(s_a, (\mathcal{H}_s^*, \mathcal{H}_e^*)) + \mathcal{L}_{\text{BCE}}(s_m, (\mathcal{H}_s^*, \mathcal{H}_e^*))))}_{\text{loss of TTS}} + \underbrace{\lambda_{\text{ASA}}(\mathcal{L}_{\text{BCE}}(c_m^*, c_m) + \mathcal{L}_{\text{BCE}}(c_a^*, c_a))}_{\text{loss of ASA}}$$
$$+ \underbrace{\lambda_k(\mathcal{L}_{\text{KL}}(\mathcal{H}_s^*, \mathcal{H}_s) + \mathcal{L}_{\text{KL}}(\mathcal{H}_e^*, \mathcal{H}_e))}_{\text{loss of temporal decoder}} + \underbrace{\lambda_l \mathcal{L}_1(\mathcal{B}^*, \mathcal{B}) + \lambda_u \mathcal{L}_{\text{IoU}}(\mathcal{B}^*, \mathcal{B})}_{\text{loss of spatial decoder}} \quad (11)$$

where $\mathcal{L}_{\text{KL}}$, $\mathcal{L}_1$, $\mathcal{L}_{\text{IoU}}$ and $\mathcal{L}_{\text{BCE}}$ are KL divergence, smooth L1, IoU, and binary cross-entropy losses.

## 4 EXPERIMENTS

**Implementation.** TA-STVG is implemented in Python using PyTorch (Paszke et al., 2019). Similar to (Gu et al., 2024a), we use pre-trained ResNet-101 (He et al., 2016) and RoBERTa-base (Liu et al., 2019) from MDETR (Kamath et al., 2021) as 2D and text backbones, and VidSwin-tiny (Liu et al., 2022) as 3D backbone. The number of attention heads is 8, and the hidden dimension of the encoder and decoder is 256. The channel dimensions $C_a$, $C_m$, $C_t$, $D$ are 2,048, 768, 768 and 256. $\delta$ and $\theta$ are empirically set to 0.5 and 0.7. We use random resized cropping as augmentation method, producing an output with a short side of 420. The video frame length $N_v$ is 64, and the text sequence length $N_t$ is 30. During training, we use Adam (Kingma & Ba, 2015) with an initial learning rate of 2e-5 for the pre-trained backbone and 3e-4 for the remaining modules (note that the 3D backbone is frozen). The loss weight parameters $\lambda_{\text{TTS}}$, $\lambda_{\text{ASA}}$, $\lambda_k$, $\lambda_l$, and $\lambda_u$ are set to 1, 1, 10, 5, and 3, respectively.

**Datasets and Metrics.** We use three datasets, *i.e.*, HCSTVG-v1/v2 (Tang et al., 2021) and Vid-STG (Zhang et al., 2020b), for experiments. HCSTVG-v1 contains 5,660 untrimmed videos, with 4,500 and 1,160 video-text pairs in training and testing sets. HCSTVG-v2 expands upon HCSTVG-v1, and comprises 10,131 training, 2,000 validation, and 4,413 testing samples. As test set annotations are not available, results are reported based on the validation set on HCSTVG-v2 as in other methods (Jin et al., 2022; Yang et al., 2022a; Lin et al., 2023b). VidSTG contains 99,943 video-text pairs. It includes 44,808 declarative and 55,135 interrogative sentences, with training, validation, and test sets containing 80,684, 8,956, and 10,303 sentences, and 5,436, 602, and 732 videos, respectively.

Following (Yang et al., 2022a), we employ m_tIoU, m_vIoU, and vIoU@R as the evaluation metrics. m_tIoU measures the ability of temporal grounding by computing the average tIoU score across all the test set. m_vIoU compares the spatial grounding performance by calculating the average of vIoU scores, and vIoU@R measures the performance using ratios of samples with vIoU greater than R in test sets. For detailed metrics, please kindly refer to (Yang et al., 2022a).

### 4.1 STATE-OF-THE-ART COMPARISON

**HCSTVG Datasets.** To validate the efficacy of our method, we compare it against other approaches on HCSTVG-v1/v2 datasets. Tab. 1 shows the comparison results on the HCSTVG-v1 test set. As shown in Tab. 1, TA-STVG achieves state-of-the-art performance on all four metrics, especially on the vIOU@0.3 metric, where it improves the score by 1.6% absolute gains compared to CG-STVG (Gu et al., 2024a). Compared to our baseline that does not employ TTS and ASA modules for

Table 1: Comparison on HCSTVG-v1 (%).

| Methods | m_tIoU | m_vIoU | vIoU@0.3 | vIoU@0.5 |
|---|---|---|---|---|
| STVGBert (Su et al., 2021) | - | 20.4 | 29.4 | 11.3 |
| TubeDETR (Yang et al., 2022a) | 43.7 | 32.4 | 49.8 | 23.5 |
| STCAT (Jin et al., 2022) | 49.4 | 35.1 | 57.7 | 30.1 |
| SGFDN (Wang et al., 2023c) | 46.9 | 35.8 | 56.3 | 37.1 |
| STVGFormer (Lin et al., 2023b) | - | 36.9 | 62.2 | 34.8 |
| CG-STVG (Gu et al., 2024a) | 52.8 | 38.4 | 61.5 | 36.3 |
| Baseline (ours) | 49.9 | 36.4 | 57.6 | 32.1 |
| TA-STVG (ours) | **53.0** (+3.1) | **39.1** (+2.7) | **63.1** (+5.5) | **36.8** (+4.7) |

Table 2: Comparison on HCSTVG-v2 (%).

| Methods | m_tIoU | m_vIoU | vIoU@0.3 | vIoU@0.5 |
|---|---|---|---|---|
| PCC (Yu et al., 2021) | - | 30.0 | - | - |
| 2D-Tan (Tan et al., 2021) | - | 30.4 | 50.4 | 18.8 |
| MMN (Wang et al., 2022) | - | 30.3 | 49.0 | 25.6 |
| TubeDETR (Yang et al., 2022a) | 53.9 | 36.4 | 58.8 | 30.6 |
| STVGFormer (Lin et al., 2023b) | 58.1 | 38.7 | 65.5 | 33.8 |
| CG-STVG (Gu et al., 2024a) | 60.0 | 39.5 | 64.5 | 36.3 |
| Baseline (ours) | 58.3 | 37.9 | 62.6 | 33.0 |
| TA-STVG (ours) | **60.4** (+2.1) | **40.2** (+2.3) | **65.8** (+3.2) | **36.7** (+3.7) |

Table 3: Comparison with existing state-of-the-art methods on VidSTG (%).

| Methods | Declarative Sentences | | | | Interrogative Sentences | | | |
|---|---|---|---|---|---|---|---|---|
| | m_tIoU | m_vIoU | vIoU@0.3 | vIoU@0.5 | m_tIoU | m_vIoU | vIoU@0.3 | vIoU@0.5 |
| STGRN (Zhang et al., 2020b) | 48.5 | 19.8 | 25.8 | 14.6 | 47.0 | 18.3 | 21.1 | 12.8 |
| OMRN (Zhang et al., 2020a) | 50.7 | 23.1 | 32.6 | 16.4 | 49.2 | 20.6 | 28.4 | 14.1 |
| STGVT (Tang et al., 2021) | - | 21.6 | 29.8 | 18.9 | - | - | - | - |
| STVGBert (Su et al., 2021) | - | 24.0 | 30.9 | 18.4 | - | 22.5 | 26.0 | 16.0 |
| TubeDETR (Yang et al., 2022a) | 48.1 | 30.4 | 42.5 | 28.2 | 46.9 | 25.7 | 35.7 | 23.2 |
| STCAT (Jin et al., 2022) | 50.8 | 33.1 | 46.2 | 32.6 | 49.7 | 28.2 | 39.2 | 26.6 |
| SGFDN (Wang et al., 2023c) | 45.1 | 28.3 | 41.7 | 29.1 | 44.8 | 25.8 | 36.9 | 23.9 |
| STVGFormer (Lin et al., 2023b) | - | 33.7 | 47.2 | 32.8 | - | 28.5 | 39.9 | 26.2 |
| CG-STVG (Gu et al., 2024a) | 51.4 | 34.0 | 47.7 | 33.1 | 49.9 | 29.0 | 40.5 | 27.5 |
| Baseline (ours) | 49.5 | 32.3 | 44.9 | 31.6 | 48.6 | 27.5 | 38.5 | 25.6 |
| TA-STVG (ours) | **51.7** (+2.2) | **34.4** (+2.1) | **48.2** (+3.3) | **33.5** (+1.9) | **50.2** (+1.6) | **29.5** (+2.0) | **41.5** (+3.0) | **28.0** (+2.4) |

target-specific query generation and instead uses zero-initialized object queries, TA-STVG shows absolute performance gains of 3.1%, 2.7%, 5.5%, and 4.7% on the four metrics, which evidences the effectiveness of our method. Tab. 2 shows the result on HCSTVG-v2 validation set. As shown, TA-STVG achieves SOTA results on all four metrics. Compared to the baseline, TA-STVG improves it with 2.1%, 2.3%, 3.2%, and 3.7% absolute gains on four metrics, validating its efficacy again.

**VidSTG Dataset.** We evaluate TA-STVG on VidSTG in Tab. 3. Different from HCSTVG, VidSTG includes both declarative and interrogative sentences. As in Tab. 3, TA-STVG achieves state-of-the-art results on all 8 metrics. using target-aware queries, TA-STVG significantly outperforms the baseline, improving m_tIoU and m_vIoU scores by 2.2% and 2.1% for declarative sentences, and 1.6% and 2.0% for interrogative sentences, respectively, once again validating the effectiveness of TA-STVG.

## 4.2 ABLATION STUDY

**Impact of TTS and ASA.** The TTS and ASA are key components of TA-STVG. To validate their effectiveness, we conducted ablation experiments on HCSTVG-v1. As shown in Tab. 4, without TTS and ASA (❶), our baseline achieves a m_tIoU score of 49.9%. When TTS is applied alone, the m_tIoU score is improved 52.2% with 2.3% gains (❶ *v.s.* ❷).

Table 4: Ablations of TTS and ASA.

| | TTS | ASA | m_tIoU | m_vIoU | vIoU@0.3 | vIoU@0.5 |
|---|---|---|---|---|---|---|
| ❶ | - | - | 49.9 | 36.4 | 57.6 | 32.1 |
| ❷ | ✓ | - | 52.2 | 38.4 | 61.7 | 36.2 |
| ❸ | - | ✓ | 51.4 | 38.0 | 60.4 | 34.1 |
| ❹ | ✓ | ✓ | **53.0** | **39.1** | **63.1** | **36.8** |

When ASA is used alone, m_tIoU is improved to 51.4% with 1.5% gains (❶ *v.s.* ❸). Working together, the best result of 53.0% is achieved with 3.1% absolute gains (❶ *v.s.* ❸). All these results clearly validate the efficacy of TTS and ASA, working either alone or together, for improving STVG.

**Impact of different branches in TTS.** In TTS, we consider both appearance and motion, guided by text, for temporal selection. To further analyze TTS, we conduct experiments on its different choices on HCSTVG-v1. The results are reported in Tab. 5. As in Tab. 5, without using TTS (but using ASA), TA-STVG achieves 51.4% m_tIoU score (❶). With the help of text-guided appearance branch, the m_tIoU score is improved to 51.8% with 0.4% gains (❶ *v.s.* ❷). When using text-guided motion branch, the m_tIoU score is improved to 52.3% with 0.9% gains (❶ *v.s.* ❸). When considering both appearance and motion, the m_tIoU score is boosted to 53.0% with 1.6% gains (❶ *v.s.* ❹), showing the best performance. It is worth noting that, if not using text as a guidance, the m_tIoU score will be decreased from 53.0% to 51.8% (❺ *v.s.* ❹), validating the importance of textual guidance in TTS.

**Impact of different attributes in ASA.** In ASA, we consider both appearance and motion attributes for fine-grained feature learning. For in-depth analysis, we conduct ablations for ASA on HCSTVG-v1. As in Tab. 6, without using ASA (but using TTS), TA-STVG achieves 52.2% m_tIoU score (❶). When considering appearance attribute, m_tIoU is improved to 52.3% (❶ *v.s.* ❷). When using motion attribute, the m_tIoU score is improved to 52.7% (❶ *v.s.* ❸). When applying both appearance and motion attributes, best result of 53.0% m_tIoU score is achieved with 0.8% gains (❶ *v.s.* ❺).

Table 5: Ablations of branches in TTS. "TG", "AB", and "MB" are the text-guided, appearance and motion branches, respectively.

| | TG | AB | MB | m_tIoU | m_vIoU | vIoU@0.3 | vIoU@0.5 |
|---|---|---|---|---|---|---|---|
| ❶ | - | - | - | 51.4 | 38.0 | 60.4 | 34.1 |
| ❷ | ✓ | ✓ | - | 51.8 | 38.4 | 61.1 | 36.3 |
| ❸ | ✓ | - | ✓ | 52.3 | 38.3 | 62.0 | 36.5 |
| ❹ | - | ✓ | ✓ | 51.8 | 38.5 | 62.0 | 36.9 |
| ❺ | ✓ | ✓ | ✓ | **53.0** | **39.1** | **63.1** | 36.8 |

Table 6: Ablations of attributes in ASA. "SG", "AA", and "MA" are the subject-guided, appearance and motion attributes, respectively.

| | SG | AA | MA | m_tIoU | m_vIoU | vIoU@0.3 | vIoU@0.5 |
|---|---|---|---|---|---|---|---|
| ❶ | - | - | - | 52.2 | 38.4 | 61.7 | 36.2 |
| ❷ | ✓ | ✓ | - | 52.3 | 38.6 | 62.4 | 36.2 |
| ❸ | ✓ | - | ✓ | 52.7 | 38.6 | 61.3 | 36.6 |
| ❹ | - | ✓ | ✓ | 52.6 | 38.8 | 61.9 | **36.8** |
| ❺ | ✓ | ✓ | ✓ | **53.0** | **39.1** | **63.1** | **36.8** |

Table 7: Ablations of activation learning in ASA.

| | Activation | m_tIoU | m_vIoU | vIoU@0.3 | vIoU@0.5 |
|---|---|---|---|---|---|
| ❶ | None | 52.2 | 38.4 | 61.7 | 36.2 |
| ❷ | Instance-act. | 52.8 | 38.8 | 62.0 | 34.7 |
| ❸ | Attribute-act. | **53.0** | **39.1** | **63.1** | **36.8** |

Table 8: Ablation of $\delta$.

| | $\delta$ | m_tIoU | m_vIoU |
|---|---|---|---|
| ❶ | 0.4 | 52.7 | 38.8 |
| ❷ | 0.5 | **53.0** | **39.1** |
| ❸ | 0.6 | 52.8 | 39.0 |

Table 9: Ablation of $\theta$.

| | $\theta$ | m_tIoU | m_vIoU |
|---|---|---|---|
| ❶ | 0.6 | 52.3 | 38.5 |
| ❷ | 0.7 | **53.0** | **39.1** |
| ❸ | 0.8 | 52.6 | 38.6 |

Please note, the gain by jointly using appearance and motion attributes is the maximum, which shows the necessity of both appearance and motion attributes for improvement. Similarly, when subject is not used, m_tIoU is decreased from 53.0% to 52.6% (❺ *v.s.* ❹), showing the need of subject in ASA.

**Analysis of activation learning strategy in ASA.** In ASA, we propose learning attribute-aware spatial activation for fine-grained target features. Another way is to learn the naive instance-level activation (not attribute-specific activation) supervised by binary masks generated from the groundtruth boxes as mentioned earlier (see these two strategies in Sec. C in supplementary material). We conduct experiments in Tab. 7. As shown, the attribute-specific features (❸) in our strategy shows better result compared to instance-level spatial activation (❷) or none activation (*i.e.*, removing ASA) (❶).

**Impact of parameter $\delta$ and $\theta$ in TTS.** In TTS, $\delta$ controls the fusion of appearance and motion scores, and $\theta$ controls the sampling of video frames. We conduct ablations in Tab. 8 and Tab. 9. We can see that the model performs best by setting $\delta$ to 0.5 (❷ in Tab. 8) and $\theta$ to 0.7 (❷ in Tab. 9).

Due to limited space, please refer to our supplementary material for more experiments and analysis.

### 4.3 VALIDATION OF GENERALITY

To validate the generality of our approach, we conduct experiments by applying TTS and ASA to other Transformer-based methods, including TubeDETR and STCAT. Since these two methods only use appearance features, we retain only the appearance branch to accommodate them. Tab. 10 reports the results on HCSTVG-v1. Please note, we retrain Tube-DETR and STCAT on our platform for fair

Table 10: Incorporate the TTS and ASA modules into different methods on HCSTVG-v1 (%). ◆: results from the original paper. ◆: retrained results.

| | Method | TTS + ASA | m_tIoU | m_vIoU | vIoU@0.3 | vIoU@0.5 |
|---|---|---|---|---|---|---|
| ❶ | TubeDETR◆ | - | 43.7 | 32.4 | 49.8 | 23.5 |
| ❷ | TubeDETR◆ | - | 43.2 | 31.6 | 49.1 | 25.5 |
| ❸ | TubeDETR◆ | ✓ | **45.5** (+2.3) | **33.5** (+1.9) | **53.0** (+3.9) | **27.1** (+1.6) |
| ❹ | STCAT◆ | - | 49.4 | 35.1 | 57.7 | 30.1 |
| ❺ | STCAT◆ | - | 48.3 | 34.9 | 57.2 | 29.8 |
| ❻ | STCAT◆ | ✓ | **50.0** (+1.7) | **36.7** (+1.8) | **59.9** (+2.7) | **31.7** (+1.9) |

comparison. Although we try our best to re-implement these methods with provided code and settings, our results differ slightly from the original paper with small discrepancy. As in Tab. 10, we observe our method effectively and consistently improves TubeDETR and STCAT, achieving gains of 2.3%/1.9% (❷ *v.s.* ❸) and 1.7%/1.8% (❺ *v.s.* ❻) in m_tIoU/m_vIoU, evidencing its generality.

## 5 CONCLUSION

This paper proposes a novel Target-Aware Transformer for STVG (TA-STVG) by exploiting target-specific cues for query generation. The key lies in two effective modules of TTS and ASA. TTS aims to sample target-relevant temporal features using text as a guidance, while ASA explores fine-grained spatial features from coarse-grained temporal features. Working in a cascade, they allow exploiting crucial target-specific cues for generating target-aware object queries, which learn better target position information for STVG. Experiments on three datasets validate the efficacy of TA-STVG.

**Acknowledgements.** Libo Zhang is supported by National Natural Science Foundation of China (No. 62476266). Yan Huang, Yuewei Lin, and Heng Fan are not supported by any funds for this work.

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

## SUPPLEMENTAL MATERIAL

For better understanding of this work, we offer additional details, analysis, and results as follow:

- **A    Details and More Results for the Oracle Experiments**
  In this section, we display more detailed architecture for the Oracle experiment and report results on more datasets.

- **B    Architectures for Self-attention Encoder, Spatial and Temporal Decoders**
  In this section, we show the details for the architecture of the self-attention encoder used for multimodal feature fusion. In addition, we display the architectures of spatial and temporal decoders.

- **C    Different Architectures of Activation Learning Strategies**
  In this section, we present details of architectures for activation learning strategies in ASA.

- **D    Text Pre-processing and Attribute Label Construction**
  This section will illustrate the necessary text pre-processing for subject extraction. In addition, we will detail the construction of vocabulary for attribute labels.

- **E    Clarification for Multi-label Classification in ASA**
  In this section, we offer clarification for the multi-label classification in the ASA module.

- **F    Additional Ablation Experiments**
  In this section, we show additional ablation experiments on different loss weights and the robustness of our method.

- **G    Additional Discussion**
  In this section, we provided additional discussions on our method.

- **H    Efficiency and Complexity Analysis**
  In this section, we provide analysis of efficiency and model complexity for our proposed method and its comparisons with other state-of-the-art approaches.

- **I    Qualitative Analysis and Results**
  In this section, we display qualitative analysis including relevance score learned by TTS, attribute-specific spatial activation learned by ASA, attention maps of object queries in the decoder, and qualitative results for spatial-temporal target localization.

- **J    Limitation**
  We discuss the limitation of our method.

## A    DETAILS AND MORE RESULTS FOR THE ORACLE EXPERIMENTS

To verify whether target-aware queries benefit localization, we conduct an Oracle experiment by generating spatial and temporal object queries from the ground-truth boxes. Similar to our TA-STVG, the architecture in this Oracle experiment consists of a multimodal encoder, query generation, and a decoder, as shown in Fig. 7. The difference with our method is that, in the Oracle experiment, object queries are directly generated from the groundtruth boxes of the target using RoI pooling. In contrast, our method generates object queries from the trained TTS and ASA modules.

We also conduct Oracle experiments on VidSTG (Zhang et al., 2020b) and HCSTVG-v2 (Tang et al., 2021) datasets, as shown in Fig. 8. The results reveal that, compared to zero-initialized queries, the groundtruth-generated queries have significantly improve performance on these two datasets, which once again demonstrates the importance of using target-aware queries in enhancing the performance of STVG.

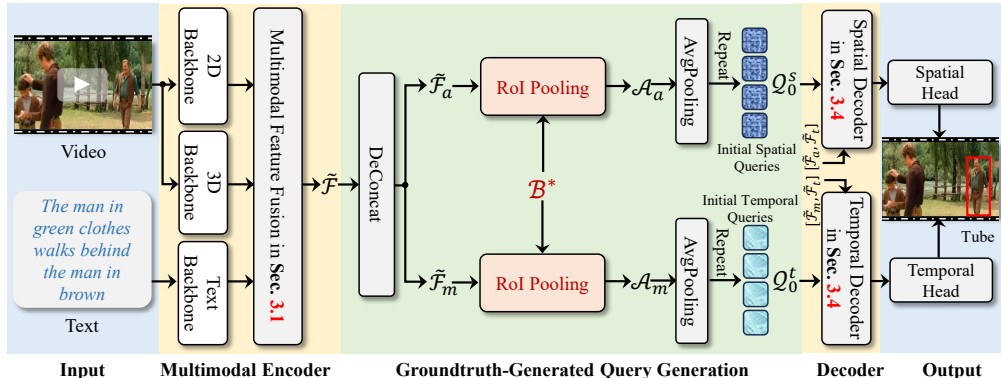

Figure 7: Detailed architecture for the oracle experiment using groundtruth-generated object queries for STVG. $\mathcal{B}^*$ denotes the ground truth bounding box used to generate object queries.

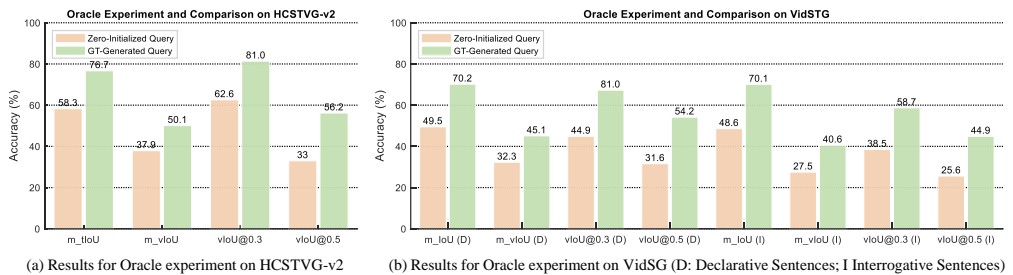

(a) Results for Oracle experiment on HCSTVG-v2     (b) Results for Oracle experiment on VidSG (D: Declarative Sentences; I Interrogative Sentences)

Figure 8: Comparison of performance using zero-initialized object queries and the groundtruth-generated object queries for STVG on HCSTVG-v2 in (a) and VidSTG in (b).

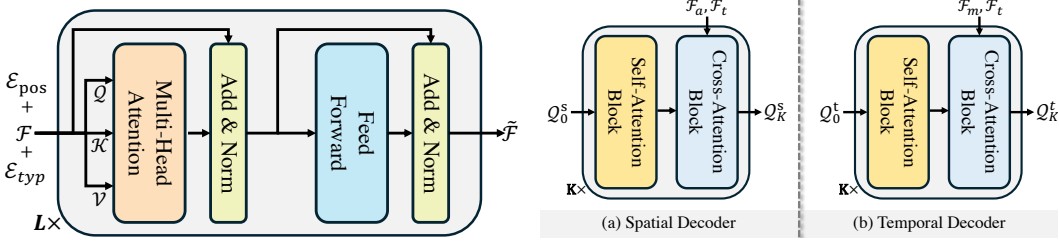

Figure 9: Architecture of self-attention encoder.      Figure 10: Architecture of decoders.

## B   ARCHITECTURES FOR SELF-ATTENTION ENCODER, SPATIAL AND TEMPORAL DECODERS

### B.1   ARCHITECTURE OF SELF-ATTENTION ENCODER

The self-attention encoder SelfAttEncoder$(\cdot)$, composed of $L$ ($L = 6$) standard self-attention encoder blocks, is used to fuse features from multiple modalities. The structure is shown in Fig. 9.

### B.2   ARCHITECTURE OF SPATIAL AND TEMPORAL DECODER

Similar to previous methods (Gu et al., 2024a; Jin et al., 2022), we use a spatial decoder to learn spatial locations and a temporal decoder to learn temporal locations. The spatial decoder and the temporal decoder each consist of $K$ ($K = 6$) blocks, with each block comprising a self-attention module and a cross-attention module. Fig. 10 illustrates the architectures of the spatial and temporal decoders.

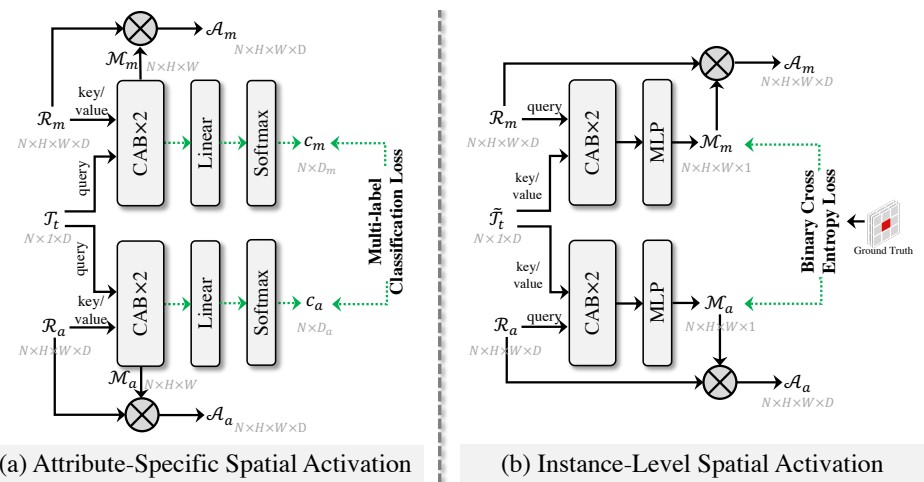

Figure 11: The architectures of attribute-specific (a) and instance-level (b) spatial activation.

## C   DIFFERENT ARCHITECTURES OF ACTIVATION LEARNING STRATEGIES

In ASA, we compare two different strategies for learning spatial activation of the target. One is the proposed attribute-aware spatial activation which learns attribute-specific information by attribute classification, and the other one is to learn the naive instance-level spatial activation through binary classification. Fig. 11 (a) and (b) show the detailed architectures of these strategies. Specifically, for the instance-level spatial activation in (b), the classification results can be directly employed as the spatial activation maps $\mathcal{M}_a$ and $\mathcal{M}_m$ to generated the spatially activated features. Compared to the naive instance-level activation method, our attribute-specific spatial activate enables learning more discriminative fine-grained features, which are crucial to distinguish the target from the background.

## D   TEXT PRE-PROCESSING AND ATTRIBUTE LABEL CONSTRUCTION

**Subject Extraction.** In the ASA module, we guide the learning of activation using the subject from the text. To this end, we employ the Stanford CoreNLP tool to conduct syntactic analysis on the text, transforming it into a structured graph. From the generated graph, we are able to extract the subject along with the related adjectives for appearance such as color and shape and verbs for motion such as various actions based on the part-of-speech tagging. The adjectives and verbs extracted are respectively used to produce the appearance and motion attribute labels. It is worth noticing that, since current STVG task focuses on localizing a single target (and thus there is only one subject in the query), we extract only one subject representation from the textual sentence.

**Vocabulary and Attribute Label Construction**. The ASA module is trained by predicting attribute classes. For this purpose, we have constructed two vocabularies for each dataset, including an appearance vocabulary and a motion vocabulary. Please note that, all information is from the dataset itself, without using extra information. Specifically, we first count the frequency of attribute words (adjectives or verbs related to the subject.) in each textual description in the dataset, and then eliminate words that appear less than 50 times. The remaining words constitute the vocabulary. The lengths of the appearance vocabulary and motion vocabulary for HCSTVG-v1/v2 are 25 and 40, respectively, while for VidSTG, they are 36 and 45, respectively. With the vocabulary, we can construct the multi-class attribute label for each dataset based on the words in the textural expression.

## E   CLARIFICATION FOR MULTI-LABEL CLASSIFICATION IN ASA

In this section, we provide more clarifications for the multi-label classification in our ASA module. Specifically, we answer the following questions: **(a)** Are the extracted subject features for appearance

---

stanfordnlp.github.io/CoreNLP/

Table 11: Ablations of different loss weights on HCSTVG-v1.

| | $\lambda_{\text{TTS}}$ | $\lambda_{\text{ASA}}$ | m_tIoU | m_vIoU | vIoU@0.3 | vIoU@0.5 |
|---|---|---|---|---|---|---|
| ❶ | 1 | 1 | **53.0** | **39.1** | 63.1 | **36.8** |
| ❷ | 1 | 5 | 52.9 | 39.0 | **63.3** | 36.5 |
| ❸ | 5 | 1 | 52.8 | 38.7 | 62.8 | 35.9 |
| ❹ | 5 | 5 | 52.6 | 38.8 | 62.7 | 36.3 |

Table 12: Comparison of existing methods in complex scenarios on Hard-HCSTVG-v1.

| | Method | m_tIoU | m_vIoU | vIoU@0.3 | vIoU@0.5 |
|---|---|---|---|---|---|
| ❶ | TubeDETR | 40.4 | 28.3 | 42.3 | 14.3 |
| ❷ | STCAT | 42.9 | 28.8 | 47.3 | 19.3 |
| ❸ | CG-STVG | 45.5 | 31.4 | 50.3 | 20.0 |
| ❹ | Baseline (ours) | 43.1 | 29.3 | 48.8 | 18.7 |
| ❺ | TA-STVG (ours) | **45.9** | **32.6** | **54.7** | **22.3** |

and motion identical? **(b)** What is the relationship between weak attribute labels and subject features, and **(c)** do the weak labels update dynamically?

**For (a)**: The extracted subject features for appearance and motion are identical. Because the subject is usually the target of interest, and we want to extract its related appearance and motion features. Thus, we use the identical subject as the query for learning both appearance and motion information. **For (b)**: The weak attribute labels and the subject both come from the given textual query (please see Sec. D in our supplementary material on how to obtain these two). For example, in the textual query 'The tall girl is walking', 'girl' is the subject, while 'tall' and 'walking' are the weak attribute labels. The subject feature is used as the query to learn relevant attribute information from the visual features for the goal of multi-attribute classification. **For (c)**: Given the textual query, the weak label for the weak attributes is generated from the query and is fixed (not updated).

## F  ADDITIONAL ABLATION EXPERIMENTS

**Impact of different loss weights.** Similarly to the previous approaches, our model is trained with multiple losses, including TTS loss, ASA loss, spatial decoder loss, and temporal decoder loss. The weights for the spatial and temporal decoder losses are kept consistent works such as STCAT (Jin et al., 2022) and CG-STVG (Gu et al., 2024a). The weights for TTS and ASA losses are empirically set. We have conducted an ablation on the weights $\lambda_{\text{TTS}}$ and $\lambda_{\text{ASA}}$ for TTS and ASA losses, and the results are shown in Tab. 11. From Tab. 11, we observe that when setting $\lambda_{\text{TTS}}$ and $\lambda_{\text{ASA}}$ to 1 and 1 (see ❶), respectively, we achieve the best performance.

**Ablation on the robustness of TA-STVG.** In order to show the robustness of our TA-STVG again challenges, we manually select videos from a current benchmark HCSTVG-v1, in which the objects suffer from heavy occlusions, similar distractors, or noisy text descriptions. The resulted subset is called **Hard-HCSTVG-v1** (it will be released together with our source code), containing 300 videos selected from the test set of HCSTVG-v1. We show our performance on this Hard-HCSTVG-v1 and compare it with other state-of-the-art models in Tab. 12. From Tab. 12, we observe that, due to increased complexity, all the models degrade on the more difficult Hard-HCSTVG-v1 (please see Tab. 12 here and Tab. 1 in the paper). Despite this, our proposed TA-STVG still achieves the best performance (45.9/32.6/54.7/22.3 in m_tIoU/m_vIoU/vIoU@0.3/vIoU@0.5, see ❺) by outperforming CG-STVG (45.5/31.4/50.3/20.0 in m_tIoU/m_vIoU/vIoU@0.3/vIoU@0.5, see ❸), STCAT (42.9/28.8/47.3/19.3 in m_tIoU/m_vIoU/vIoU@0.3/vIoU@0.5, see ❷), and TubeDETR (40.4/28.3/42.3/14.3 in m_tIoU/m_vIoU/vIoU@0.3/vIoU@0.5, see ❶). Further more, TA-STVG significantly improves its baseline under these challenges (❺ *v.s.* ❹). All of these are attributed to our TTS and ASA modules that mine discriminative information of the target, even in challenging scenarios, for localization, showing their efficacy for improving STVG.

## G  ADDITIONAL DISCUSSION

**Discussion on construction of frame-specific object queries.** In this work, the feature output by the ASA module undergoes a pooling and repetition to generate the final object queries. The ***reasons***

for not constructing frame-specific object queries are two-fold: (1) For STVG, we aim to obtain a global-level target-relevant query by pooling, which ensures the consistency in target prediction across different frames, *i.e.*, localizing the same target instance in all frames using global information; and (2) Since the ASA module only extracts target features from frames selected by the previous TTS module, not every frame has its own target-specific feature. As a result, it is hard to directly construct frame-specific object queries in our work.

**Discussion of computational consumption and future directions**. Our TA-STVG needs a high demand of computational resource. In fact, this is a common issue in the STVG field. The computational complexity of STVG methods is high, because STVG models, particularly current Transformer-based ones (including our work), require to receive all sampled video frames at once to make the prediction, leading to the requirement of significant computational resources. That being said, our proposed modules TTS and ASA are lightweight and only bring about 4M parameters and 0.09T FLOPS compared to the baseline method (please see Tab. 13). In the future, we will further explore efficient STVG in the future, such as (1) how to develop lightweight STVG architecture by reducing unnecessary network parameters, aiming to improve training and inference efficiency, and (2) how to leverage parameter-efficient fine-tuning (PETR) techniques like adapter or prompt learning for STVG model training, aiming to reduce the training complexity.

**Discussion on handling videos with multiple targets**. For STVG, the video may comprise multiple targets which are described in a single textual query. Since the target of interest is often the unique subject of the textual query, our model is able to leverage many descriptions such as motion and color attributes via our proposed modules, as well as contextual information like interaction with others by our spatio-temporal modeling in TA-STVG to distinguish the target from other objects.

**Discussion on adapting TTS and ASA to other multimodal data.** In this work, we introduce two modules TTS and ASA for STVG. Besides STVG, TTS and ASA (with appropriate adaptations) can be applied to work with other multimodal data. The essence of both TTS and ASA is to align two different correlated modalities (in our work, they are visual and text modalities). Specifically, TTS aims to align the frame-level feature with the global sentence-level feature, while ASA works to align the region-level feature with the local word-level feature. For other multimodal data, such as audio description and video, TTS and ASA can be used in a similar way. For example, TTS can be used to align the video with the global audio description to remove irrelevant video content, and ASA can be applied to align spatial regions in the video to more fine-grained local audio descriptions (like attributes in textual description). For 3D videos, TTS and ASA can be used similarly.

**Analysis on multi-label classification in the ASA**. In this work, the extracted subject features for appearance and motion are identical. Because the subject is usually the target of interest, and we want to extract its related appearance and motion features. Thus, we use the identical subject as the query for learning both appearance and motion information. we use the identical subject as the query for learning both appearance and motion information. The attribute labels and the subject both come from the given textual query. Given the textual query, the label for the weak attributes is generated from the query, and is fixed. The subject feature is used as the query to learn relevant attribute information from the visual features for the goal of multi-attribute classification.

**Discussion on sampled target-relevant features on ASA.** In our TA-STVG, the information of sampled frames by TTS is crucial for the subsequent ASA (which extracts attribute-related target information from sample frames for generating queries). We discuss the impact of sampled target-relevant frames on ASA in two situations: **(a)** Part of the selected frames does not contain the target object, while the other part contains the target. In this case, since the attention mechanism is utilized in the subsequence ASA, it can still extract useful target-aware information from partial sampled frames that contain the target while decreasing the weight of sampled frames without targets. As a result, we can still generate effective target-aware query for improving STVG. For the worse case in this situation, the subsequent ASA fails, the sampled frames from TTS alone, even only partial frames contain targets, can still generate useful queries to enhance STVG performance, as shown in Tab. 4 (see ❶ and ❷ in Tab. 4 in the paper). **(b)** All sampled frames from TTS do not contain the target. If unfortunately this happens, then the final generated queries after ASA will not contain effective target-aware information, which is equivalent to using randomness-initialized query for STVG and thus may degrade the final performance. However, we think that this case might be *very occasional*, as from Tab. 4 in the paper, we can see that TTS alone can significantly improve baseline

Table 13: Comparison on model efficacy and complexity on VidSTG. The inference is conducted on a single A100 GPU, and inference time refers to the duration of a single forward propagation.

| Methods | Params | | Training | | Inference | | FLOPS | m_tIoU | m_vIoU |
|---|---|---|---|---|---|---|---|---|---|
| | Trainable | Total | Time | GPU Num | Time | GPU Mem | | | |
| TubeDETR (Yang et al., 2022a) | 185M | 185M | 48 h | 16 V100 | 0.40 s | 24.4 | 1.45 T | 48.1 | 30.4 |
| STCAT (Jin et al., 2022) | 207M | 207M | 12 h | 32 A100 | 0.51 s | 30.4 | 2.85 T | 50.8 | 33.1 |
| STVGFormer (Lin et al., 2023b) | - | - | ∼48 h | 8 A6000 | - | - | - | - | 33.7 |
| CG-STVG (Gu et al., 2024a) | 203M | 231M | 13.6 h | 32 A100 | 0.61 s | 29.7 | 3.03 T | 51.4 | 34.0 |
| Baseline (ours) | 202M | 230M | 14 h | 32 A100 | 0.53 s | 28.2 | 2.88 T | 49.5 | 32.3 |
| TA-STVG (ours) | 206M | 234M | 14.5 h | 32 A100 | 0.57 s | 28.4 | 2.97 T | **51.7** | **34.4** |

performance. This means that TTS can usually generate effective sampled frames that contain the target. We may leave the exploration of this to our future work.

**Discussion on our innovation and contribution.** In this work, we emphasize that, our innovation and contribution is the *idea* of *exploring target-aware queries for improving STVG*, which has *never* been studied in the STVG field. To achieve the target-aware queries for STVG, we proposed two modules, TTS and ASA. The reason why implementing TTS and ASA with attention is because the adopted cross-attention perfectly meets our demand in extracting object query features and is simply enough (the pursuit of *simple but effective* architecture is always our motivation). In experiments, we show our proposed TA-STVG is able to achieve the best performance on all three datasets, showing the significant contribution of our idea of exploring target-aware for STVG. Moreover, our idea is general and when applied to other frameworks such as STCAT and TubeDETR, we demonstrate consistent improvements, evidencing its efficacy.

## H  EFFICIENCY AND COMPLEXITY ANALYSIS

To analyze the efficacy and complexity of our model, we conduct a comparative analysis involving the number of parameters, training consumption, inference consumption, and FLOPS against other existing methods. As shown in Tab. 13, we can see that, our proposed maintains similar computational complexity (206/234M trainable/total parameters and 2.97T FLOPS) to existing state-of-the-art models such as STCAT (207M/207M trainable/total parameters and 2.85T FLOPS) and CG-STVG (203M/231M trainable/total parameters and 3.03T FLOPS), while achieving better performance with the best m_tIoU of 51.7 and m_vIoU of 34.4 on VidSTG. Moreover, compared to the baseline method, TA-STVG brings negligible parameter and computational increments (Baseline *v.s.* TA-STVG: 202M *v.s.* 206M in trainable parameters, 230M *v.s.* 234M in total parameters, and 2.88T *v.s.* 2.97T in FLOPS), yet significantly enhances the performance (m_tIoU improved from 49.5 to 51.7 in and m_vIoU from 32.3 to 34.4), underscoring its superiority.

## I  QUALITATIVE ANALYSIS AND RESULTS

### I.1  VISUALIZATION OF TEMPORAL RELEVANCE SCORE AND ATTRIBUTE-SPECIFIC ACTIVATION

In this section, we show more visualizations of the temporal relevance scores learned by TTS and attribute-specific spatial activation learned by ASA in Fig. 12. From Fig. 12, we can see that the frames, selected by the temporal relevance score $s$, will be within the groundtruth, which shows the selected features are target-relevant. In addition, the attribute-specific spatial attention maps $\mathcal{M}_a$ and $\mathcal{M}_m$ can well localize the related attributes such as color and different actions of the target. F instance, in the first example, the text is "The *man* in *green* clothes *walks* behind the man in brown". The highly relevant areas in the appearance attention map are mainly concentrated around the green clothes, while in the motion attention map are focused on the legs. These attribute-specific spatial activation helps with learning more discriminative fine-grained features for improving STVG.

### I.2  ATTENTION MAPS OF OBJECT QUERIES IN THE SPATIAL DECODER

To show the role of target-aware queries, we visualize its attention maps in the spatial decoder with comparison to the attention maps of zero-initialized queries, as shown in Fig. 13. As in Fig. 13, the

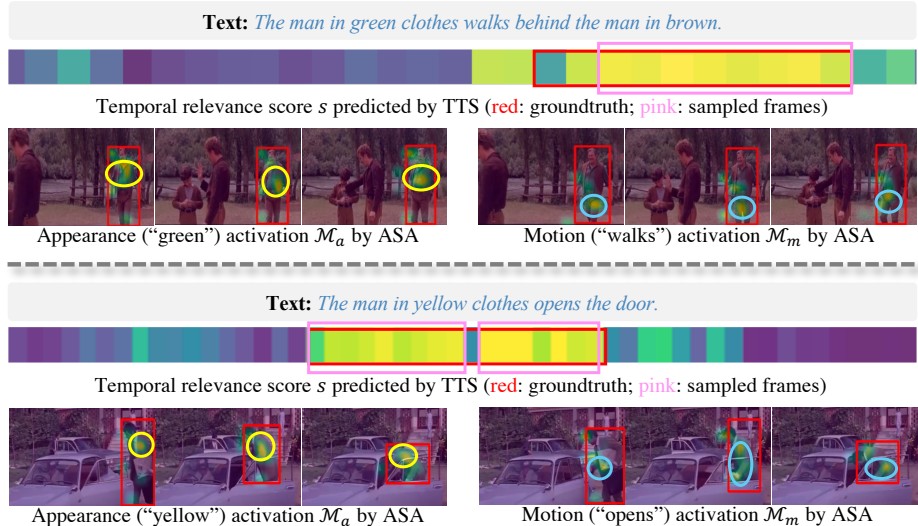

Figure 12: Visualization of temporal relevance score predicted by TTS and attribute-aware activation score predicted by ASA. The red boxes indicate the temporal and spatial foreground object to localize.

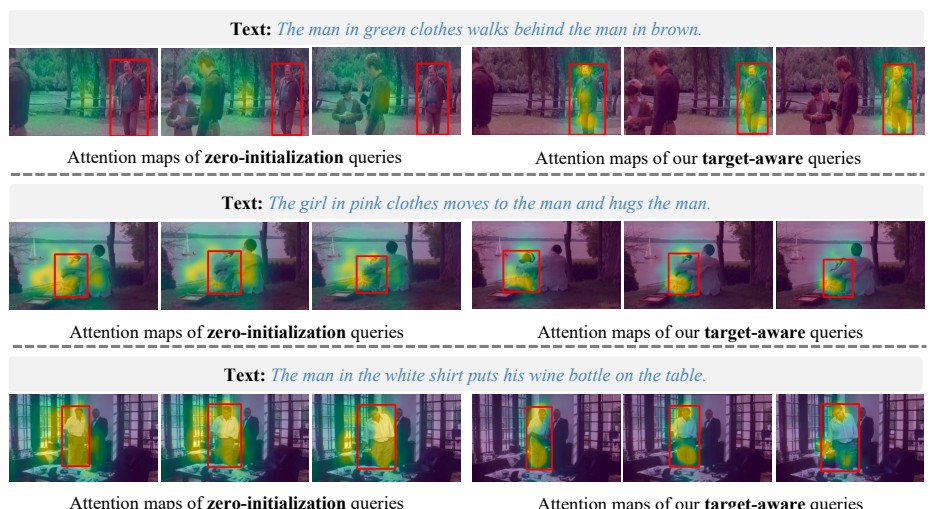

Figure 13: Comparison of attention maps for zero-initialized object queries and the proposed target-aware object queries in video frames in the spatial decoder. The red boxes indicate the foreground target to localize. From this figure, we can clearly observe that the attention maps of our target-aware object queries by TTS and ASA can better focus on the target object for localization.

attention maps from zero-initialized queries are scattered, leading to imprecise target localization. In contrast, the attention maps from target-aware queries highly focus on the target, demonstrating that target-aware object queries can better learn target position information for improving STVG.

## I.3 QUALITATIVE GROUNDING RESULTS

To further qualitatively validate the effectiveness of our proposed method, we provide its grounding results and comparison with the baseline method (with zero-initialization queries) on HCSTVG-v1/v2, as shown in Fig. 14. From Fig. 14, it is evident that our method outperforms the baseline in both temporal and spatial localization. This suggests that queries with target-specific fine-grained visual information can better learn the spatio-temporal position of the target, thereby enhancing spatio-temporal grounding capability of the model.

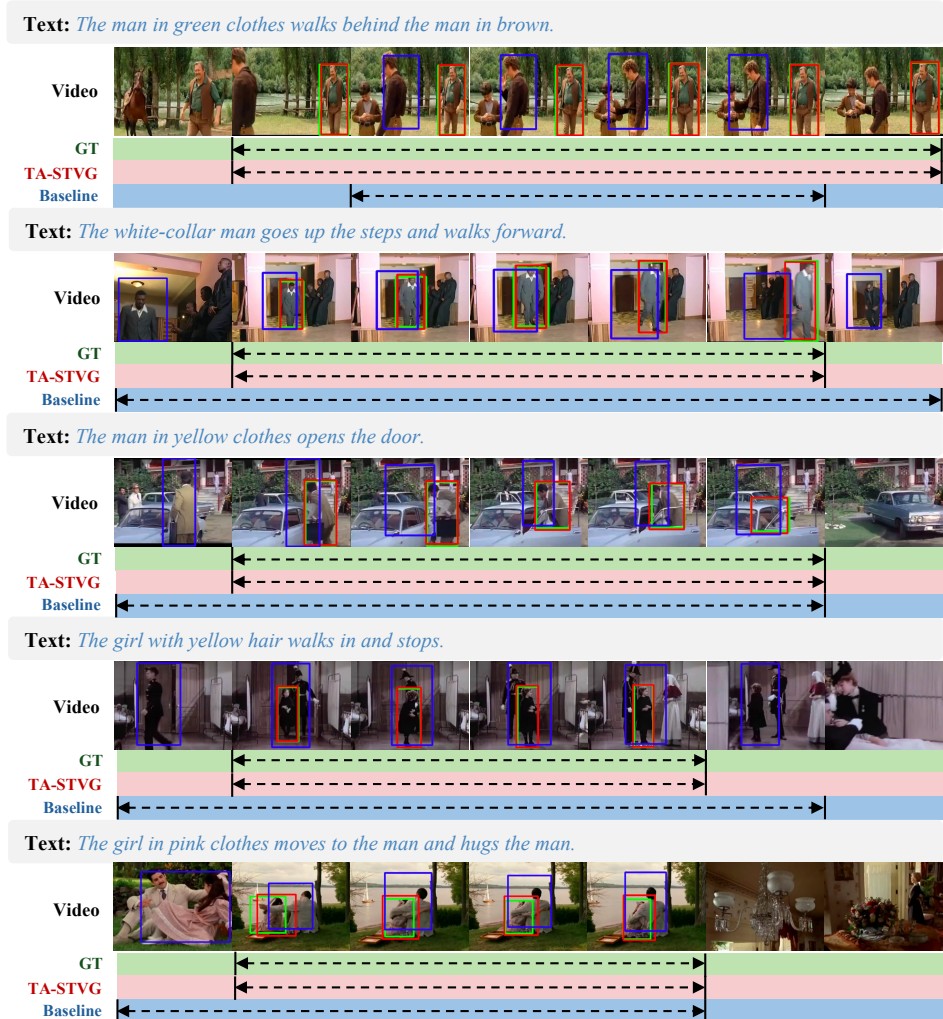

Figure 14: Qualitative results of our TA-STVG (red), the baseline (blue), and ground truth (green).

For instance, in the first example, the task is to locate a man wearing green clothes and walking. The baseline model, using zero-initialized queries, fails to locate the target due to incorrect identification. In contrast, our model, through the proposed TTS and ASA modules, extracts target-related visual information such as "green" and "walk" from the video as initial queries, accurately locating the target object. The same comparison is observed in other examples. These results and comparison further show that initializing queries with fine-grained visual information of target can improve STVG.

## J    LIMITATION

Despite achieving state-of-the-art performance on multiple datasets, our method has two main limitations. First, although TTS and ASA are able to provide better initialization for object queries, these queries still inevitably contain some noise compared to the desired information, resulting in lower performance than the Oracle experiments with groundtruth-generated object queries. In the future, one direction is to further improve the quality of our target-aware object queries. Second, similar to all other STVG approaches, our method consumes a large amount of computational resources for training, leading to lengthy training period. In future, we will study resource-friendly STVG using techniques from parameter-efficient learning methods.

