# OpenReview forum: "Knowing Your Target: Target-Aware Transformer Makes Better Spatio-Temporal Video Grounding"
_ICLR.cc/2025/Conference — ICLR 2025 Oral_

### Official Review · Reviewer_oPic · 2024-10-31

**Soundness:** 3
**Presentation:** 3
**Contribution:** 3
**Rating:** 8
**Confidence:** 4

**Summary:**

This paper addresses the challenge of spatio-temporal video grounding (STVG) in complex scenarios by introducing the Target-Aware Transformer for STVG (TA-STVG). TA-STVG adaptively generates object queries based on target-specific cues from the video-text pair, utilizing two key modules: text-guided temporal sampling (TTS) and attribute-aware spatial activation (ASA). TTS selects relevant temporal cues using text information, while ASA refines these cues to capture detailed visual attributes for initializing object queries. Experiments on benchmarks, including HCSTVG-v1/-v2 and VidSTG, demonstrate that TA-STVG achieves state-of-the-art performance.

**Strengths:**

- Unlike previous methods that use zero-initialized object queries, this paper presents the Target-Aware Transformer, which improves STVG by exploring target-specific cues for generating object queries.
- This paper proposes text-guided temporal sampling to select target-relevant temporal cues from videos and introduces attribute-aware spatial activation to leverage fine-grained visual semantic attributes for object query generation.
- Experiments on three datasets demonstrate its effectiveness.

**Weaknesses:**

- Given the textual feature, first extract the subject's feature (usually the target to localize) from the textual sentence. My question is, do you typically obtain just one subject representation or multiple? If multiple are obtained, are they all processed through the same flow for decoding, or is there a further selection step?
- Sampled target-relevant temporal appearance and motion features—do they need to have the same quantity? If so, how is this controlled?
- The sampled target-relevant temporal appearance and motion features determine the frame information for the target object. This step can significantly impact subsequent accuracy. What happens if the selected frames do not contain the target object?

**Questions:**

- Given the textual feature, first extract the subject's feature (usually the target to localize) from the textual sentence. My question is, do you typically obtain just one subject representation or multiple? If multiple are obtained, are they all processed through the same flow for decoding, or is there a further selection step?
- Sampled target-relevant temporal appearance and motion features—do they need to have the same quantity? If so, how is this controlled?
- The sampled target-relevant temporal appearance and motion features determine the frame information for the target object. This step can significantly impact subsequent accuracy. What happens if the selected frames do not contain the target object?

---

> ### Author Response · Authors · 2024-11-23
> **Rebuttal**
>
> We appreciate the reviewer for careful comments and provide our responses below.
>
> > **Q1**: Given the textual feature, first extract the subject's feature (usually the target to localize) from the textual sentence. My question is, do you typically obtain just one subject representation or multiple? If multiple are obtained, are they all processed through the same flow for decoding, or is there a further selection step?
>
> **A1**: Thanks for this comment. Since **current STVG focuses on localizing a single target** (and thus there is only one subject in the query), we extract only one subject representation from the textual sentence.
>
> To make this point clear, we have included the above clarification in the revised version (`please see L846-848 of Sec. D in the supplementary material`). Thanks again!
>
> > **Q2**: Sampled target-relevant temporal appearance and motion features, do they need to have the same quantity? If so, how is this controlled?
>
>
> **A2**: Thanks for this careful comment. For the sampled target-relevant temporal appearance and motion features, *i.e.*, $\\mathcal{R}\_a$ and $\\mathcal{R}\_m$, their quantities are the same. We set the same threshold $\theta$ for both appearance and motion features, and sample $\\mathcal{R}\_a$ and $\\mathcal{R}\_m$ from $\\mathcal{\\tilde{F}}\_{a}$ and $\\mathcal{\\tilde{F}}\_{m}$ ($\\mathcal{\\tilde{F}}\_{a}$ and $\\mathcal{\\tilde{F}}\_{m}$ have the same quantities) both based on the final relevance score $s$  generated by TTS (*please see Eq. (6) in the paper*). Therefore, their quantities are the same, and consistent with the number of frames sampled by TTS.
>
> To clarify this point, we have added the above explanation in the revised version (`please see L298-L301 in the revised paper`). Thanks again!
>
> > **Q3**: The sampled target-relevant temporal appearance and motion features determine the frame information for the target object. This step can significantly impact subsequent accuracy. What happens if the selected frames do not contain the target object?
>
> **A3**: Thanks and we really like this insightful and interesting comment. Yes, the sampled frame information by TTS is crucial for the subsequent ASA (which extracts attribute-related target information from sample frames for generating queries). We discuss the point mentioned by the reviewer in two situations:
> **(a)** **Part** of the selected frames do not contain the target object, while the other part contains the target. In this case, since the attention mechanism is utilized in the subsequence ASA, it can still extract useful target-aware information from partial sampled frames that contain the target while decreasing the weight of sampled frames without targets. As a result, we can still generate effective target-aware query for improving STVG. For the worse case in this situation, the subsequent ASA fails, the sampled frames from TTS alone, even only partial frames contain targets, can still generate useful queries to enhance STVG performance, as shown in Tab. 4 (see ➊ and ➋ in Tab. 4 in the paper).
> **(b)** **All** sampled frames from TTS do not contain the target. If unfortunately this happens, then the final generated queries after ASA will not contain effective target-aware information, which is equivalent to using randomness-initialized query for STVG and thus may degrade the final performance. However, we think that this case might be **very occasional**, as from Tab. 4 in the paper, we can see that TTS alone can significantly improve baseline performance. This means that TTS usually can generate effective sampled frames that contain the target. We may leave the exploration of this to our future work.
>
> We thank the review again for this insightful and interesting comment, which can allow readers to better understand our method. We have included the above analysis in the revised version to make this point clearer (`please see L958-L985 of Sec. G in the supplementary material`).
>
> Thanks again!

---

### Official Review · Reviewer_g9Sv · 2024-11-02

**Soundness:** 3
**Presentation:** 3
**Contribution:** 3
**Rating:** 6
**Confidence:** 4

**Summary:**

The paper introduces a Target-Aware Transformer to address the limitations of traditional zero-initialized queries in STVG. The proposed method leverages target-specific information to generate more adaptive object queries that better interact with multimodal features, leading to improved accuracy. Extensive experiments on multiple benchmarks demonstrate that TA-STVG achieves SOTA performance.

**Strengths:**

1. Target-specific cues improve object queries over zero-init methods, enhancing feature learning for better localization.

2. Extensive ablation studies support the effectiveness of the TTS and ASA modules, and the generality of modules is further validated by the integration into other models (e.g. TubeDETR, STCAT).

**Weaknesses:**

1. The method’s contribution is focused primarily on object query initialization for STVG, with the innovation mainly revolving around attention-based filtering for keyframes and target-specific attribute extraction, which makes the innovation limited.

2. The multi-label classification in the ASA module is unclear. Specifically, are the extracted subject features for appearance and motion identical? What is the relationship between weak attribute labels and subject features, and do the weak labels update dynamically?

3. The performance in complex scenarios requires further validation. Since object query initialization relies on the provided video-text pair, how robust is the query generation when visual information is heavily occluded, or the query is vague and lacks clear attributes? The robustness of the model against unclear attribute information should be further explored.

**Questions:**

See weakness

---

> ### Author Response · Authors · 2024-11-23
> **Rebuttal (part 1/2)**
>
> We thank the reviewer for careful comments on our work and provide our responses below.
>
> > **Q1**: The method’s contribution is focused primarily on object query initialization for STVG, with the innovation mainly revolving around attention-based filtering for keyframes and target-specific attribute extraction, which makes the innovation limited.
>
> **A1**: Thanks for this careful comment. Here, we'd like to stress that, our innovation is the the **idea** of **exploring target-aware queries for improving STVG**, which has **never** been studied in the STVG field and thus is recognized by reviewer **CJ8J** ("*intuitive motivation of TA-STVG is clear*") and reviewer **jmHQ** ("*a novel approach to STVG*", "*significant departure from traditional zero-initialized queries*"). To achieve the target-aware queries for STVG, we proposed two modules, TTS and ASA. The reason why implementing TTS and ASA with attention is because the adopted cross-attention perfectly meets our demand in extracting object query features and is simply enough (the pursuit of **simple but effective** architecture is always our motivation). In experiments, we show our proposed TA-STVG is able to achieve the best performance on all three datasets, showing the significant contribution of our idea ("*exploring target-aware for STVG*"). Moreover, our idea is general and when applied to other frameworks such as STCAT and TubeDETR, we demonstrate consistent improvements, evidencing its efficacy.
>
> We have included the above clarification in revision to make our innovations and contributions clearer (`please see L985-L993 in Sec. G of the supplementary material`). Again, thanks!
>
> > **Q2**: The multi-label classification in the ASA module is unclear. Specifically, **(a)** are the extracted subject features for appearance and motion identical? **(b)** What is the relationship between weak attribute labels and subject features, and **(c)** do the weak labels update dynamically?
>
>
> **A2**: Sorry for the confusion. To better answer the reviewer's concern, we break down the question into three smaller questions **(a)**, **(b)**, and **(c)**. **For (a)**, yes, the extracted subject features for appearance and motion are identical. Because the subject is usually the target of interest, and we want to extract its related appearance and motion features. Thus, we use the identical subject as the query for learning both appearance and motion information. **For (b)**, the weak attribute labels and the subject both come from the given textual query (please see Sec. D in our supplementary material on how to obtain these two). For example, in the textual query ‘*The tall girl is walking*’, ‘*girl*’ is the subject, while ‘*tall*’ and ‘*walking*’ are the weak attribute labels. The subject feature is used as the query to learn relevant attribute information from the visual features for the goal of multi-attribute classification. **For (c)**, given the textual query, the weak label for the weak attributes is generated from the query, and is fixed (not updated).
>
> We thank the reviewer again for these careful comments, and we have inlcuded the above clarification in the revised version (`please see L861-L888 in Sec. E of the supplementary material`). Thanks!

---

> > ### Author Response · Authors · 2024-11-23
> > **Rebuttal (part 2/2)**
> >
> > > **Q3**: The performance in complex scenarios requires further validation. Since object query initialization relies on the provided video-text pair, how robust is the query generation when visual information is heavily occluded, or the query is vague and lacks clear attributes? The robustness of the model against unclear attribute information should be further explored.
> >
> > **Tab. D: Comparison of existing methods in complex scenarios on Hard-HCSTVG-v1.**
> >
> > || Method            | m\_tIoU | m\_vIoU | vIoU\@0.3 | vIoU\@0.5 |
> > |---|-------------------|:---------:|:---------:|:----------:|:----------:|
> > |➊| TubeDETR          | 40.4    | 28.3    | 42.3     | 14.3     |
> > |➋| STCAT             | 42.9    | 28.8    | 47.3     | 19.3     |
> > |➌| CG-STVG           | 45.5    | 31.4    | 50.3     | 20.0     |
> > |➍| Baseline (ours)   | 43.1    | 29.3    | 48.8     | 18.7     |
> > |➎| TA-STVG (ours)    | **45.9**| **32.6**| **54.7** | **22.3** |
> >
> > **A3**: Thanks and we really like this thoughtful comment. In order to show the robustness of our TA-STVG again challenges mentioned by the reviewer, we manually select the videos from a current benchmark HCSTVG-v1, in which the objects suffer from heavy occlusions, similar distractors, or noisy text descriptions. The resulted subset is called **Hard-HCSTVG-v1** (it will be released together with our source code), containing 300 videos selected from the test set of HCSTVG-v1. We show our performance on this Hard-HCSTVG-v1 and compare it with other state-of-the-art models, in Tab. D. From Tab. D, we see that, due to increased complexity, all the models degrade on the more difficult Hard-HCSTVG-v1 (please see Tab. D here and Tab. 1 in the paper). Despite this, our proposed TA-STVG still achieves the best performance (45.9/32.6/54.7/22.3 in m\_tIoU/m\_vIoU/vIoU\@0.3/vIoU\@0.5, see ➎) by outperforming CG-STVG (45.5/31.4/50.3/20.0 in m\_tIoU/m\_vIoU/vIoU\@0.3/vIoU\@0.5, see ➌), STCAT (42.9/28.8/47.3/19.3 in m\_tIoU/m\_vIoU/vIoU\@0.3/vIoU\@0.5, see ➋), and TubeDETR (40.4/28.3/42.3/14.3 in m\_tIoU/m\_vIoU/vIoU\@0.3/vIoU\@0.5, see ➊). In addition, TA-STVG significantly improves its baseline under these challenges (➎ *v.s.* ➍). All these are attributed to our TTS and ASA modules that mine discriminative information of the target, even in challenging scenarios, for localization, showing their efficacy for improving STVG.
> >
> > We thank the reviewer again, and have integrated the above results and analysis in the revised version (`please see L900-L912 in Sec. F of the supplementary material`).

---

> > > ### Comment · Reviewer_g9Sv · 2024-11-27
> > > **About the rebuttal**
> > >
> > > Thanks for providing clarification and more experimental results. I have read all the other reviews and the authors' rebuttal, I believe the paper has good quality, and the response has addressed my concerns. Considering the combination properties of TTS and ASA modules for STVG task, I maintain the initial score.

---

> > > > ### Author Response · Authors · 2024-11-27
> > > >
> > > > Thank you for your positive feedback and for the time and effort you have dedicated to reviewing our work. We are glad our response addressed your concerns. Thank you again for your sincere dedication.

---

### Official Review · Reviewer_jmHQ · 2024-11-02

**Soundness:** 3
**Presentation:** 3
**Contribution:** 3
**Rating:** 8
**Confidence:** 3

**Summary:**

The paper introduces an approach to spatio-temporal video grounding (STVG) using a Target-Aware Transformer (TA-STVG). The method aims to improve the localization of targets within videos based on textual descriptions by generating object queries that are adaptively initialized with target-specific cues extracted from video-text pairs. The key contributions include two main modules: Text-Guided Temporal Sampling (TTS) for selecting target-relevant frames, and Attribute-Aware Spatial Activation (ASA) for exploiting fine-grained visual attribute information. The proposed method achieves state-of-the-art performance on three benchmarks: HCSTVG-v1/-v2 and VidSTG, and demonstrates generality by improving the performance of existing Transformer-based methods like TubeDETR and STCAT.

**Strengths:**

1. The paper presents a novel approach to STVG by introducing target-aware object queries, which is a significant departure from traditional zero-initialized queries.

2. TA-STVG outperforms existing methods on multiple benchmarks, indicating the effectiveness of the proposed approach.

3. The writing of the paper is good and it is easy to understand.

**Weaknesses:**

1.The method may require significant computational resources, which may limit its accessibility and practical application, especially for smaller research groups or companies.

2. The paper could provide more discussion on the computational complexity and parameter count compared to other methods, which is crucial for understanding its scalability.

3. The paper could benefit from a discussion on the robustness of the model against various challenges such as occlusion, similar appearances, or noisy text descriptions.

**Questions:**

1. How does the computational complexity of TA-STVG compare to other state-of-the-art methods in terms of parameters and FLOPS?


2. How does the model handle videos with multiple targets that are described in a single text query?

3. Can the TTS and ASA modules be adapted to work with other types of multimodal data, such as audio descriptions or 3D video data?

---

> ### Author Response · Authors · 2024-11-23
> **Rebuttal (part 1/2)**
>
> We thank the reviewer for his careful and helpful comments on our work. We provide our responses below to answer the reviewer's questions.
>
> > **Q1**: The method may require significant computational resources, which may limit its accessibility and practical application, especially for smaller research groups or companies.
>
> **A1**: Thanks for this insightful comment. We understand the reviewer's concern regarding the high demand for the computational resource in our work. In fact, this is a **common issue** to the STVG field. The computational complexity of STVG methods is high, because STVG models, particularly current Transformer-based ones (including our work), need to take as input **all** sampled video frames **at once** to make the prediction, leading to the requirement of significant computational resources. That being said, our proposed modules TTS and ASA are lightweight and only bring around 4M parameters and 0.09T FLOPS compared to the baseline method (please see Tab. B later).
>
> We are thankful to the reviewer for this insightful comment. The reviewer's comment will motivate us to further explore efficient STVG in the future, such as *(1) how to develop lightweight STVG architecture by reducing unnecessary network parameters*, aiming to improve training and inference efficiency and *(2) how to leverage parameter-efficient fine-tuning (PETR) techniques like adapter or prompt learning for STVG model training*, aiming to reduce the training complexity, just to name a few.
>
> Thanks again for this helpful comment! We have included the above analysis in the revised version (`please see L924-L934 in the Sec. G of the superiority material`).
>
> > **Q2**: The paper could provide more discussion on the computational complexity and parameter count compared to other methods, which is crucial for understanding its scalability.
>
> **Tab. B: Comparison of computational complexity.**
> ||        Methods          | Trainable Params       | &nbsp; Total Params &nbsp;      | &nbsp;&nbsp;FLOPS&nbsp;&nbsp;  | &nbsp;&nbsp;m\_tIoU&nbsp;&nbsp;    | &nbsp;&nbsp;m\_vIoU&nbsp;&nbsp; |
> |---|-----------------------|:----------------------:|:------------------:|:------:|:----------:|:-------:|
> |➊| TubeDETR                | 185M                   | 185M               | 1.45T  | 48.1       | 30.4    |
> |➋| STCAT                   | 207M                   | 207M               | 2.85T  | 50.8       | 33.1    |
> |➌| CG-STVG                 | 203M                   | 231M               | 3.03T  | 51.4       | 34.0    |
> |➍| Baseline (Ours)         | 202M                   | 230M               | 2.88T  | 49.5       | 32.3    |
> |➎| TA-STVG (Ours)          | 206M                   | 234M               | 2.97T  | **51.7**   | **34.4**|
>
>
> **A2**: Thanks for this constructive comment. In our supplementary material, we have compared our TA-STVG with other state-of-the-art methods regarding computational complexity (please kindly see Tab. 13 in the supplementary material). For the reviewer's convenience, we copy the information about the comparison of computational complexity (including the number of parameters and FLOPS) from Tab. 13 here in Tab. B. From Tab. B, we can see that, our proposed maintains similar computational complexity (206/234M trainable/total parameters and 2.97T FLOPS, see ➎) to existing state-of-the-art models such as STCAT (207M/207M trainable/total parameters and 2.85T FLOPS, see ➋) and CG-STVG (203M/231M trainable/total parameters and 3.03T FLOPS, see ➌), while achieving better performance with the best m\_tIoU of 51.7 and m\_vIoU of 34.4 on VidSTG. Moreover, compared to the baseline method (➍), TA-STVG brings negligible parameter and computational increments (Baseline *v.s.* TA-STVG: 202M *v.s.* 206M in trainable parameters, 230M *v.s.* 234M in total parameters, and 2.88T *v.s.* 2.97T in FLOPS), yet significantly enhances the performance (m\_tIoU improved from 49.5 to 51.7 in and m\_vIoU from 32.3 to 34.4), underscoring its superiority.
>
> We appreciate the reviewer for this helpful comment and have included the above analysis and discussion in the revised version (`please see L999-L1007 in Sec. H of the supplementary material`). Thanks again!

---

> ### Author Response · Authors · 2024-11-23
> **Rebuttal (part 2/2)**
>
> > **Q3**: The paper could benefit from a discussion on the robustness of the model against various challenges such as occlusion, similar appearances, or noisy text descriptions.
>
> **Tab. C: Comparison of existing methods in complex scenarios on Hard-HCSTVG-v1.**
>
> || Method            | m\_tIoU | m\_vIoU | vIoU\@0.3 | vIoU\@0.5 |
> |---|-------------------|:---------:|:---------:|:----------:|:----------:|
> |➊| TubeDETR          | 40.4    | 28.3    | 42.3     | 14.3     |
> |➋| STCAT             | 42.9    | 28.8    | 47.3     | 19.3     |
> |➌| CG-STVG           | 45.5    | 31.4    | 50.3     | 20.0     |
> |➍| Baseline (ours)   | 43.1    | 29.3    | 48.8     | 18.7     |
> |➎| TA-STVG (ours)    | **45.9**| **32.6**| **54.7** | **22.3** |
>
> **A3**: Thanks and we really like this insightful comment. In order to show the robustness of our TA-STVG again challenges mentioned by the reviewer, we manually select videos from a current benchmark HCSTVG-v1, in which the objects suffer from heavy occlusions, similar distractors, or noisy text descriptions. The resulted subset is called **Hard-HCSTVG-v1** (it will be released together with our source code), containing 300 videos selected from the test set of HCSTVG-v1. We show our performance on this Hard-HCSTVG-v1 and compare it with other state-of-the-art models, in Tab. C. From Tab. C, we see that, due to increased complexity, all the models degrade on the more difficult Hard-HCSTVG-v1 (please see Tab. C here and Tab. 1 in the paper). Despite this, our proposed TA-STVG still achieves the best performance (45.9/32.6/54.7/22.3 in m\_tIoU/m\_vIoU/vIoU\@0.3/vIoU\@0.5, see ➎) by outperforming CG-STVG (45.5/31.4/50.3/20.0 in m\_tIoU/m\_vIoU/vIoU\@0.3/vIoU\@0.5, see ➌), STCAT (42.9/28.8/47.3/19.3 in m\_tIoU/m\_vIoU/vIoU\@0.3/vIoU\@0.5, see ➋), and TubeDETR (40.4/28.3/42.3/14.3 in m\_tIoU/m\_vIoU/vIoU\@0.3/vIoU\@0.5, see ➊). In addition, TA-STVG significantly improves its baseline under these challenges (➎ *v.s.* ➍). All these are attributed to our TTS and ASA modules that mine discriminative information of the target, even in challenging scenarios, for localization, showing their efficacy for improving STVG.
>
> As suggested, we have integrated the above results and analysis in the revised version (`please see L900-L912 in Sec. F of the supplementary material`). Thanks again!
>
>
> > **Q4**: How does the computational complexity of TA-STVG compare to other state-of-the-art methods in terms of the parameters and FLOPS?
>
> **A4**: Thanks for this comment. Please kindly refer to our **A2** to your **Q2** for the comparison of parameters and FLOPS.
>
> > **Q5**: How does the model handle videos with multiple targets that are described in a single text query?
>
> **A5**: Thanks for this careful comment. Since the **STVG task** itself aims to **localize a single target** based on the textual query, we think the reviewer is wondering about how our method can deal with locating a single specific target from multiple objects given a single textual query (please let us know if our understanding of your question is inaccurate). Since the target of interest is often the unique subject of the textual query, our model is able to leverage many descriptions such as motion and color attributes via our proposed modules, as well as contextual information like interaction with others by our spatio-temporal modeling in TA-STVG to distinguish the target from other objects.
>
> To make this point clearer, we have added the above clarification in the revised version (`please see L935-L939 in Sec. G of the supplementary material`). Thanks again!
>
> > **Q6**: Can the TTS and ASA modules be adapted to work with other types of multimodal data, such as audio descriptions or 3D video data?
>
> **A6**: Thanks for this insightful comment. We believe that TTS and ASA (with appropriate adaptations) can be applied to work with other multimodal data. The essence of both TTS and ASA is to align two different correlated modalities (in our work, they are visual and text modalities). Specifically, TTS aims to align the frame-level feature with the global sentence-level feature, while ASA works to align the region-level feature with the local word-level feature. For other multimodal data, such as audio description and video, TTS and ASA can be utilized in a similar way. For example, TTS can be used to align the video with the global audio description to remove irrelevant video content, and ASA can be applied to align the spatial regions in the video to more fine-grained local audio descriptions (like attributes in the textual description). For 3D videos, TTS and ASA can be used similarly.
>
> Again, thanks for this insightful comment, and we have included the above analysis in our revised version (`please see L940-L949 in Sec. G of the supplementary material`).

---

> > ### Comment · Reviewer_jmHQ · 2024-11-27
> >
> > After reading all the other reviews and the authors' rebuttal, I believe the paper has good quality and the authors addressed my concerns. I will raise my score.

---

> > > ### Author Response · Authors · 2024-11-27
> > >
> > > We are happy to hear that our response addressed your concerns. We sincerely appreciate the time and effort you have dedicated to reviewing our work. Once again, thank you.

---

### Official Review · Reviewer_CJ8J · 2024-11-03

**Soundness:** 4
**Presentation:** 4
**Contribution:** 3
**Rating:** 8
**Confidence:** 5

**Summary:**

In this work, a new method called Target-Aware Transformer for STVG (TA-STVG) is proposed for addressing the Spatio-Temporal Video Grounding (STVG) task. To replace the traditional zero-initialized object queries used in previous methods, TA-STVG is designed to generate target-specific object queries from the given video-text pair, with two main modules named Text-guided Temporal Sampling (TTS) and Attribute-aware Spatial Activation (ASA). Experiments and analysis are conducted to validate the effectiveness of the proposed TA-STVG.

**Strengths:**

1.The intuitive motivation of TA-STVG is clear and easy to understand, i.e., the adaptively generated object queries containing target-relevant information facilitate the spatio-temporal localization of textual expressions.

2.The proposed method to enhance and keep track of the information about the described subject in the sentence is concise and effective.

3.State-of-the-art performances are achieved by the method across all three datasets.

4.Qualitative results and analysis are adequately given to illustrate the effectiveness of the proposed components.

**Weaknesses:**

1.It would be better to include more discussions and comparisons on some recent works in the related domain of video grounding. For example, the proposed ASA module adopts similar techniques of spatial activation learning [1] and concept classification [2].

2.In the proposed TA-STVG, object queries are generated from globally pooling the multimodal features and are repeatedly used for each frame in the video. This way of object query construction seems suboptimal since the frame-specific information is ignored for the queries. Is there any consideration or motivation for such design choice? Why not construct the frame-specific object queries?

3.How are the weights for different loss terms determined? Are there any ablation results?

4.Some typos exist in the current manuscript, for example, "such" should have been "such as " in Line 29.

References [1] Collaborative static and dynamic vision-language streams for spatio-temporal video grounding. In CVPR 2023. [2] Siamese Learning with Joint Alignment and Regression for Weakly-Supervised Video Paragraph Grounding. In CVPR 2024.

**Questions:**

Please refer to the weaknesses.

**Details Of Ethics Concerns:**

None.

---

> ### Author Response · Authors · 2024-11-23
> **Rebuttal**
>
> We thank the reviewer for helpful comments on our work and provide our responses to the reviewer's questions below.
>
> > **Q1**: It would be better to include more discussions and comparisons on some recent works in the related domain of video grounding. For example, the proposed ASA module adopts similar techniques of spatial activation learning [1] and concept classification [2].
> [1] Collaborative static and dynamic vision-language streams for spatio-temporal video grounding. In CVPR 2023.
> [2] Siamese Learning with Joint Alignment and Regression for Weakly-Supervised Video Paragraph Grounding. In CVPR 2024.
>
> **A1**: Thanks for this helpful comment. We agree with the reviewer that our work is related to the mentioned papers [1, 2]. Specifically, the method of [1] explores activating the spatially attended target using intra-frame visual cues, and the work of [2] studies how to jointly learn feature alignment and regression using weak supervision signals, both to some extent inspiring our work.
>
> As suggested, we have included the above discussion and cite these two references in revision (`please see L113-L116 in the revised paper`). Thanks again!
>
>
> > **Q2**: In the proposed TA-STVG, object queries are generated from globally pooling the multimodal features and are repeatedly used for each frame in the video. This way of object query construction seems suboptimal since the frame-specific information is ignored for the queries. Is there any consideration or motivation for such design choice? Why not construct the frame-specific object queries?
>
> **A2**: Thanks for this insightful comment. In this work, the feature output by the ASA module undergoes a pooling and repetition to generate the final object queries. The **reasons** for not constructing frame-specific object queries are two-fold: **(1)** For STVG, we aim to obtain a global-level target-relevant query by pooling, which ensures the consistency in target prediction across different frames, *i.e.*, localizing the same target instance in all frames using global information; and **(2)** Since the ASA module only extracts target features from frames selected by the previous TTS module, not every frame has its own target-specific feature. As a result, it is hard to directly construct frame-specific object queries in our work.
>
> However, we really like the reviewer's point and believe that the point mentioned by the reviewer is worth exploring and has potential in an appropriately designed framework in the future.
>
> We have included the above analysis and discussion in the revision (`please see L917-L923 in Sec. G of the supplementary material`). Again, thanks!
>
> > **Q3**: How are the weights for different loss terms determined? Are there any ablation results?
>
> **Tab. A: Ablations of different loss weights on HCSTVG-v1.**
> ||$\lambda_{\textbf{TTS}}$ | $\lambda_{\textbf{ASA}}$ | **m\_tIoU** | **m\_vIoU** | **vIoU\@0.3** | **vIoU\@0.5**
> |:---:|:------------------------:|:------------------------:|-------------:|-------------:|:--------------:|:--------------:|
> |➊|1                      | 1                      | **53.0**    | **39.1**        | 63.1         | **36.8**
> |➋|1                      | 5                      | 52.9        | 39.0        | **63.3**     | 36.5
> |➌|5                      | 1                      | 52.8        | 38.7        | 62.8         | 35.9
> |➍|5                      | 5                      | 52.6        | 38.8        | 62.7         | 36.3
>
>
> **A3**: Thanks for this comment. Similarly to previous work, our model is trained with multiple losses, including TTS loss, ASA loss, spatial decoder loss, and temporal decoder loss. The weights for the spatial and temporal decoder losses are kept consistent works such as STCAT (Jin et al.) and CG-STVG (Gu et al.). The weights for TTS and ASA losses are empirically set. As suggested by the reviewer, we have conducted an ablation on the weights $\lambda_{TTS}$ and $\lambda_{ASA}$ for TTS and ASA losses, and the results are shown in Tab. A. From Tab. A, we observe that when setting $\lambda_{TTS}$ and $\lambda_{ASA}$ to 1 and 1 (see ➊), respectively, we achieve the best performance.
>
>
> We thank the reviewer again for this comment and have included the discussion of the above and the results in the revised version (`please see L893-L898 in Sec. F of the supplementary material`).
>
>
> > **Q4**: Some typos exist in the current manuscript, for example, "such" should have been "such as " in Line 29.
>
> **A4**: Thanks for pointing this out. We have double-checked and polished our manuscript to fix the typos. Thanks again!

---

> > ### Comment · Reviewer_CJ8J · 2024-11-26
> >
> > After reading all the other reviews and the authors' rebuttal, I think my concerns are well addressed and this work has solid quality and contributions in tackling one of the important vision-language tasks STVG, so I am willing to further raise my score to 8.

---

> > > ### Author Response · Authors · 2024-11-26
> > >
> > > Dear Reviewer,
> > > We sincerely appreciate the time and effort you have dedicated to reviewing our work. Your insightful comments and constructive feedback are highly valued. Again, thank you very much.

---

### Author Response · Authors · 2024-11-25
**General Rebuttal**

We appreciate all the reviewers for their thoughtful and careful feedback. In this paper, we propose a novel Target-Aware Transformer for STVG (TA-STVG) that uses target-specific cues for query generation. Experiments on three datasets validate its efficacy in improving target localization. Additionally, applying it to other frameworks like STCAT and TubeDETR shows consistent improvements, demonstrating its generalizability.

As suggested by the reviewers, we have thoroughly revised our manuscript and address each of the issues raised in the reviews:

- Reviewer `CJ8J`:  We have discussed the missing related works and will cite them in revision (*Q1*), further explained the motivation behind the query construction method (*Q2*), and included ablation experiments on loss weights (*Q3*). We also polished the whole paper to fix typos (*Q4*).

- Reviewer `jmHQ`: We have explained the computational requirements of our method (*Q1*), provided a comparison of computational complexity with other methods (*Q2*, *Q4*), and discussed the robustness of the model (*Q3*). We further analyzed handling multiple targets in a single text query (*Q5*) and the adaptability of TTS and ASA modules (*Q6*).

- Reviewer `g9Sv`: We have clarified our novelty and core contributions (*Q1*), explained the multi-label classification in the ASA module (*Q2*), and further validated the performance in complex scenarios (*Q3*).

- Reviewer `oPic`: We have clarified the details of extracting subjects from the text (*Q1*), the number of target-relevant features sampled by TTS (*Q2*), and further discussed what happens if the selected frames do not contain the target object (*Q3*).

We once again express our heartfelt gratitude to all the reviewers for their valuable feedback, and we hope that our responses satisfactorily address all concerns. Please feel free to let us know if you have any remaining concerns and we are happy to address them!

---

### Meta-Review · Area_Chair_ACJH · 2024-12-16

**Metareview:**

All reviewers recognize the novelty and contributions of this paper. The paper presents a novel approach to STVG by introducing target-aware object queries, which is a significant departure from traditional zero-initialized queries.The motivation is clear and easy to understand. After rebuttal, the authors have addressed the reviewer's main concerns. So, this paper is recommended for acceptance.

**Additional Comments On Reviewer Discussion:**

Reviewer comments include missing comparison with recent work, ablation experiments of loss functions, model parameters and complexity, and the role of the motivation. The authors have solved the above problems well.

---

### Decision · Program_Chairs · 2025-01-22

Accept (Oral)